# CD4+T-cells create a stable mechanical environment for force-sensitive TCR:pMHC interactions

Lukas Schrangl [1,2], Florian Kellner[3,9], René Platzer [3,10], Vanessa Mühlgrabner[3], Paul Hubinger[3], Josephine Wieland[4], Reinhard Obst[4], José L. Toca-Herrera [1], Johannes B. Huppa [3,5,6,7], Gerhard J. Schütz [2] & Janett Göhring[3,8] ✉

Mechanical forces acting on ligand-engaged T-cell receptors (TCR) have previously been implicated in T-cell antigen recognition and ligand discrimination, yet their magnitude, frequency, and impact remain unclear. Here, we quantitatively assess forces across various TCR:pMHC pairs with different bond lifetimes at single-molecule resolution, both before and during T-cell activation, on platforms that either include or exclude tangential force registration. For this purpose, we use glass-supported lipid bilayers presenting pMHC conjugated to a molecular force sensor unit at its base, adhesion factors and costimulatory molecules to the approaching T-cells. Our results imply that CD4 + T-cell TCRs experience significantly lower forces than previously estimated, with only a small fraction of ligand-engaged TCRs being subjected to these forces during antigen scanning. These rare and minute mechanical forces do not impact the global lifetime distribution of the TCR:ligand bond. We propose that the immunological synapse is created as biophysically stable environment to prevent pulling forces from disturbing antigen recognition.

The adaptive leg of immune surveillance is realized by the interplay between antigen-presenting cells (APC) and lymphocytes. Once primed in secondary lymphoid organs, T-cells become highly motile to patrol tissues in search of pathogen-derived antigens presented on the cell surface of APCs by major histocompatibility complexes (MHC)[1]. When measured in vitro, T-cell receptor (TCR) interactions with nominal peptide/MHCs (pMHC) are typically of moderate affinity. Yet, T-cells can detect the presence of even a single antigenic pMHC among thousands of non-stimulatory but structurally similar ligands[2,3]. Upon encountering antigen, T-cells rapidly initiate via their triggered TCRs intracellular signaling cascades and establish, through a complex interplay of TCRs,

accessory molecules, and the underlying cytoskeleton, an elaborate bi-membrane T-cell:APC interface, termed the immunological synapse[4]. Its unique properties and temporal plasticity are likely to influence the dynamics of receptor-ligand interactions[5]. For example, massive cytoskeletal rearrangements[5] after initial contact eventually result in the molecular segregation of membrane receptors and ligands based on the size of their extracellular domains[6]. Furthermore, during activation TCRs are transported from the periphery to the center of the synapse[7,8], potentially applying tension on the TCR:pMHC bond[9–11].

While intrinsic properties of the receptor-ligand bond appear to only partially determine the signaling outcome, biophysical

[1]Department of Bionanosciences, Institute of Biophysics, University of Natural Resources and Life Sciences, Vienna, Austria. [2]Institute of Applied Physics, TU Wien, Vienna, Austria. [3]Institute for Hygiene and Applied Immunology, Center for Pathophysiology, Infectiology and Immunology, Medical University of Vienna, Vienna, Austria. [4]Institute for Immunology, Biomedical Center, Medical Faculty, Ludwig-Maximilians-Universität München, Planegg-Martinsried, München, Germany. [5]Institute of Cancer Immunology, Charité - Universitätsmedizin, Berlin, Germany. [6] Max Delbrück Center, Berlin, Germany. [7]German Cancer Consortium (DKTK), Heidelberg, Germany. [8]Department of Biotechnology and Food Science, Institute of Molecular Biotechnology, University of Natural Resources and Life Sciences, Vienna, Austria. [9]Present address: Valdospan, Tulln, Austria. [10]Present address: Institute for Pharmacology, Johannes Kepler University Linz, Linz, Austria. ✉e-mail: janett.goehring@meduniwien.ac.at

parameters such as mechanical forces have also been implicated[12–19]. Exerting defined mechanical tension onto the TCR:pMHC bond revealed that molecular force indeed can activate T-cells[13,14] and even improve ligand discrimination[20–25] by increasing (catch bond) or decreasing (slip bond) binding lifetimes[26–28]. Catch bond behavior was reported with a maximum lifetime increase at 10-15 pN per bond for CD8 + [20,21,23,24,29] and 15–17 pN per bond for CD4 + T-cells[25]. Another study involving MHC class II molecules as ligands for CD8 + T-cells implied dynamic catch bonding for the bimolecular complex[22]. TCRs of CD8 + T-cells experience a force range of 12–19 pN[30] or >4.7 pN[31] per bond within the immunological synapse as quantified using DNA-based molecular force sensors. In contrast, the quantification of molecular forces exerted by the TCR itself revealed that CD4 + T-cells do not necessarily reach the peak force regime of the dynamic catch bond[11,32]. Furthermore, when assessing TCR:pMHC interactions in a cell-free context to preclude contextual accessory interactions, catch bond formation was no longer observed[33]. Taken together, the physiological significance of mechanical forces exerted and experienced by pMHC-engaged TCRs remains unclear: molecular tension could be a supportive or perturbing factor or even both in T-cell antigen recognition[34]. The principle of "force-shielding", i.e. the protection of TCR:ligand pairs from molecular tension by surrounding adhesion molecules such as CD2 and LFA-1 has been suggested in order to create a stable biophysical environment[34,35].

Here, we quantified mechanical forces as they were exerted by single pMHC-engaged TCRs within the immunological synapse of CD4 + T-cells prior to and during T-cell activation. We next correlated observed forces with synaptic lifetimes of the respective TCR:pMHC pairs to assess the possibility of catch–slip bond formation. To this end, we employed single-molecule-sensitive microscopy modalities monitoring the optical parameters of (i) a peptide-based molecular force sensor attached to the base of the ligand of interest and anchored to a surrogate APC, or (ii) a proximity-sensitive sensor to measure lifetimes of receptor-ligand interactions. In this fashion, we succeeded in determining TCR-exerted molecular forces as well as their frequency of occurrence in synapses of CD4 + T-cells, their impact on TCR:pMHC lifetimes, and ultimately their relevance for T-cell antigen recognition and activation.

We find that only a small proportion of TCR:pMHC bonds are subjected to discernible forces, and force magnitudes are substantially lower than expected from external pulling experiments. Additionally, comparing scenarios of different force amplitudes, we observe no effect on the TCR:pMHC bond lifetime distribution. We therefore conclude that, within our experimental settings, mechanical forces are prevented rather than employed in CD4 + T-cells' immunological synapse formation and maintenance.

## Results

### Quantification of TCR-imposed molecular forces with a Molecular Force Sensor platform

To quantitate molecular TCR-imposed forces as they occur within the immunological synapse, we built a molecular force sensor (MFS) platform as previously described[11,36] (see Fig. 1 for a schematic overview). Briefly, glass-supported lipid bilayers (SLB) were decorated with murine ICAM-1 and B7-1, and with monovalent streptavidin serving as an anchor unit for biotinylated MFSs. The MFS platform supports direct visualization via total internal reflection fluorescence (TIRF) microscopy and quantification of TCR-exerted forces within the single-digit piconewton (pN) range via single-molecule FRET. Two fluorophores constituting a FRET pair were attached to the peptide backbone derived from the flagelliform spider silk protein. The peptide acted as an entropic spring which collapsed in the absence of force giving rise to high FRET efficiency, while force application increased the inter-dye distance resulting in reduced FRET yield. The relation between FRET efficiency and applied force is

well-known (see Methods). The distal part of the peptide was attached to the base of a TCR ligand such as the MHC class II molecule IE$^k$ loaded with moth cytochrome C (MCC) or a high affinity derivate of MCC (affinity-enhanced peptide, AEP)[37]. Both functional units represent stimulating antigens for 5c.c7 and AND TCR-transgenic CD4 + T-cells, which were investigated in this study. Importantly, the AND TCR possesses a higher affinity for IE$^k$/MCC than 5c.c7 TCR[38]. Sensor integrity, mobility, the stoichiometry of the platform, and the usability of these constructs for single-molecule force analysis have been shown in detail before[11] or have been determined accordingly (Supplementary Fig. 1).

The SLB-based system (Fig. 1a) allowed for modulation of lateral ligand mobility by varying the lipid composition, resulting in gel-phase or fluid SLBs, and the density and strength of the provided antigenic stimulus determined the level of T-cell activation. On fluid-phase SLBs, MFSs diffuse freely with a diffusion constant $D$ of about 0.7 µm²/s[11], which is similar to the majority fraction (60–80%) of pMHC molecules at the APC plasma membrane ($D$ between 0.24 and 0.63 µm²/s)[39]. Due to the low viscosity of fluid-phase SLBs, pulling forces can be applied only perpendicularly to the membrane. MFSs bound to gel-phase SLBs are essentially immobilized[11] and serve as a model of the minority fraction (20–40%) of pMHCs on APC membranes, which exhibit confined diffusion (radius<75nm)[39]. The high SLB viscosity permits both perpendicular and tangential forces[11].

SLBs equipped with MFSs at low density (fewer than 0.1/µm²), suitable for single-molecule experiments, prompted T-cells to scan for antigen without activation (therefore referred to as "scanning conditions"), while the addition of high-density (50–100 molecules per µm²) unlabeled constructs (MFS$_0$-IE$^k$/MCC) stimulated the T-cells to establish immunological synapses (hence termed "activating conditions"; Fig. 1b). Activation status of T-cells was verified by calcium imaging employing Fura-2. We additionally noted that scanning conditions generally result in motile cells with changing cellular footprints, while activating conditions lead to rounded, stationary footprints. Both scanning and activating conditions are highly relevant regarding antigen recognition: The former induce T-cells to search for cognate pMHC, while the latter lead to the formation and maintenance of immunological synapses, which can persist for hours depending on continuous signaling induced by agonist pMHC[40].

Single-molecule time traces of FRET donor and acceptor molecules were recorded employing alternating laser excitation[41] in total internal reflection fluorescence (TIRF) microscopy and subsequently filtered and analyzed[11,42]. To determine the proportion of sensors experiencing forces ("high-force fraction"), first a FRET efficiency threshold was derived from data recorded in the absence of T-cells and hence without force (see the Methods section for details). Data points with FRET efficiency below this threshold (note that low efficiency corresponds to high force) were counted among the high-force fraction (see Fig. 1c, left column). For quantification of the sensors subjected to forces, we estimated their force probability density functions (PDF) as follows: The FRET efficiency PDF from sensors recorded in the absence of T-cells was scaled and subtracted from the efficiency PDF from sensors imaged within areas of T-cell:SLB contact (Supplementary Fig. 2, see the methods section for details). The resulting efficiency PDF was transformed to a PDF of forces using the sensors' calibration parameters (Fig. 1c, right column). From this, summary statistics, in particular median and inter-quartile range, were extracted.

As previously noted[11], noise is a limiting factor in our single-molecule measurements. Taking representatively the activating IE$^k$/MCC-functionalized MFSs on fluid-phase bilayers (about 5000 data points), we find that in the absence of cells (i.e., all sensors are collapsed), the FRET efficiencies appear normally distributed with $\mu = 0.865 \pm 0.005$ and $\sigma = 0.164 \pm 0.004$. The uncertainty

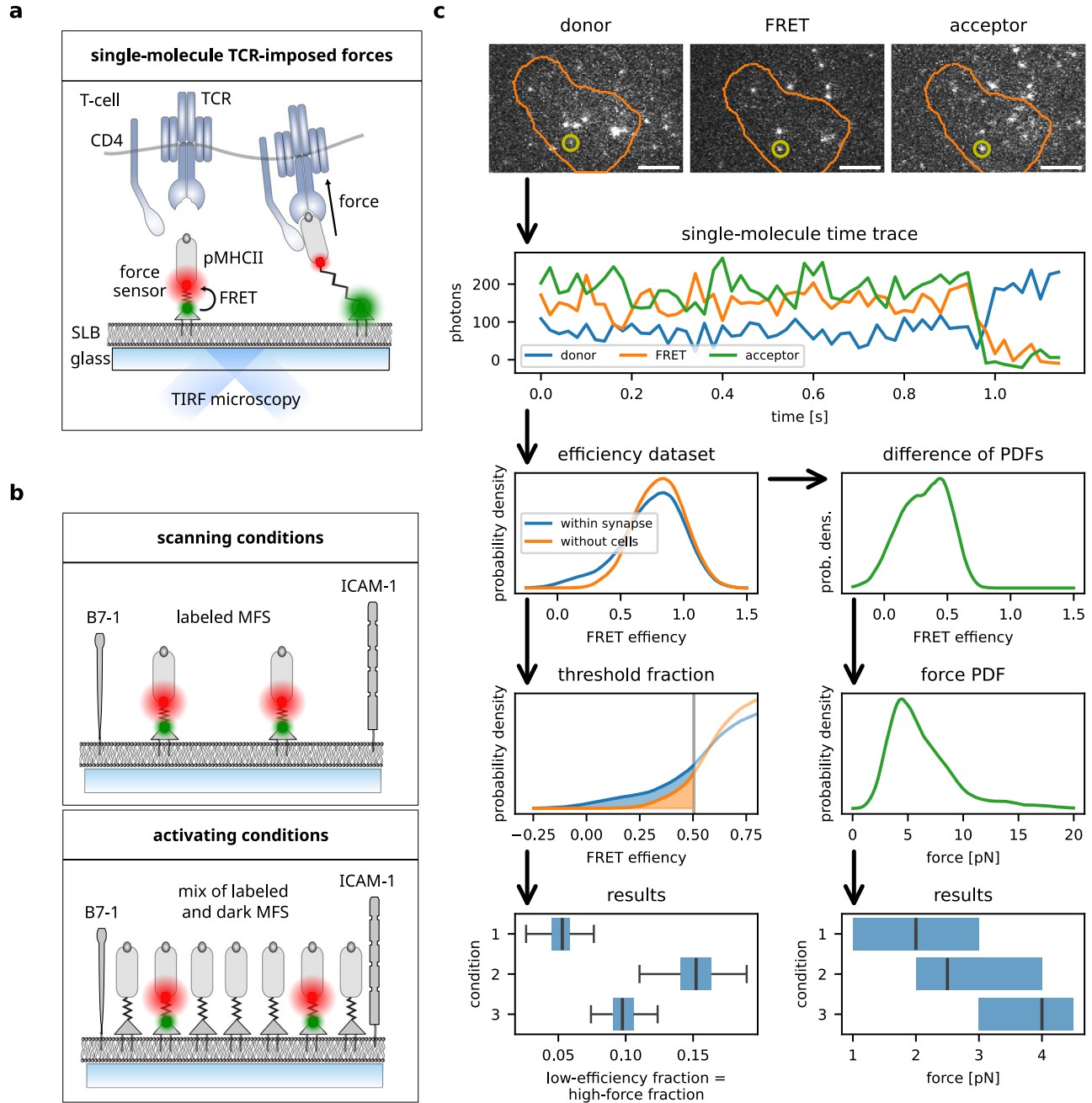

**Fig. 1 | Single-molecule force measurements to quantify TCR-imposed mechanical forces. a** Schematic representation of analog peptide-based molecular force sensor (MFS) platform. His-tagged monovalent streptavidin is directly incorporated into Ni-NTA-DGS-doped supported lipid bilayer (SLB) which can either be fluid or in a gel phase. MFS-conjugated pMHC of interest is anchored via biotin-streptavidin interaction. Single-molecule microscopy (total internal reflection) allows for the measurement of the distance of two fluorophores attached to the peptide backbone of the MFS via Förster resonance energy transfer (FRET). The SLB is also decorated with adhesion (ICAM-1) and costimulatory (B7-1) molecules to promote T-cell adhesion and activation. TCR-transgenic T-cells interact with the MFS-coupled pMHC and exert molecular forces extending the spring element of the MFS and increasing the distance between the attached FRET fluorophores, which leads to a decrease in FRET efficiency. Note that forces are not necessarily directed perpendicularly to the SLB, especially if sensors are immobilized.

Depending on the TCR complex's resistance to tilting, this may influence the shear and bending stress the TCR:pMHC bond is subjected to. **b** Schematic representation of activating and scanning conditions. SLBs used for scanning conditions are decorated with fluorescence-labeled MFS at low density, whereas stimulatory SLBs are additionally decorated with non-fluorescent MFS. **c** Schematic representation of the data analysis pipeline from data acquisition of single-molecule FRET trajectories to force histograms and statistical data handling. Single-molecule FRET trajectories are transformed into a FRET efficiency histogram. Events with a FRET efficiency below the threshold defined via a 5% false positive rate in data in the absence of cells are counted to compute the proportion of high-force events (box-and-whisker plots, left column). For quantification of single-molecule force, the sample and no-cell PDFs are subtracted, and the resulting distribution of the low FRET events are converted into force and displayed as box plots (right column). Scale bar: 5 μm.

margin $\Delta\mu := 0.005$ of $\mu$ estimates the inherent uncertainty in our system. This maps to an uncertainty of less than 0.15 pN for tension of up to 9 pN applied to the MFS, which may, however, be amplified by subsequent analysis steps.

## Binding affinity does not correlate with TCR-imposed mechanical forces

We chose well-characterized TCR:pMHC pairs of different affinities in order to investigate the impact of TCR-exerted tension. AND and

5c.c7 TCR-transgenic CD4 + T-cells were confronted with IE$^k$/MCC-functionalized MFS. Of note, the AND TCR had been reported to exhibit a three times higher 2D affinity towards IE$^k$/MCC compared to the 5c.c7 TCR[38]. We also monitored forces resulting from the interaction between 5c.c7 TCRs and IE$^k$/AEP which, based on surface plasmon resonance recordings, feature a 20-fold increased 3D affinity compared to 5c.c7:IE$^k$/MCC[37]. The structural integrity of the newly synthesized MFS-IE$^k$/AEP sensor was characterized via size exclusion chromatography (Supplementary Fig. 1a–d). Its ability to efficiently bind and activate T-cells was confirmed via calcium flux analysis (Supplementary Fig. 1e). T-cell's ability for ligand discrimination was verified for 37 °C and room temperature (Supplementary Fig. 1f).

Forces were quantified both for scanning and activating conditions (see Fig. 2a, b for a summary of results, Supplementary Figs. 2 and 3 for detailed plots of distributions, and Supplementary Table 1 for summary statistics). Comparing TCR-imposed forces we observe that AND TCRs exerted larger pulling forces than the 5c.c7 TCRs regardless of experimental settings (Fig. 2a). For 5c.c7 cells which had been confronted with gel-phase SLBs under scanning conditions or alternatively with fluid-phase SLBs under activating conditions, FRET-based force measurements within synapses or in the absence of T-cells did not give rise to any significant differences (adapted KS-test[43], see the Methods section), and thus we abstained from further analysis. Surprisingly, the high-affinity 5c.c7:IE$^k$/AEP pair experienced even smaller forces than the 5c.c7:IE$^k$/MCC pair. Only under activating conditions on gel-phase SLBs we observed statistically significant deviations from the baseline. As a control, we performed experiments employing MFSs equipped with H57 TCRβ-reactive single-chain antibody fragments (scF$_V$), which exhibit very long lifetimes at room temperature with 50% receptor:ligand bonds still intact after three hours and longer (Supplementary Fig. 4a). The forces measured were similar to those recorded with AND T-cells confronted with IE$^k$/MCC ligand.

We hence conclude that TCR-exerted mechanical forces within the synaptic environment can reach 6–7 pN (3$^{rd}$ quartile) for the high-affinity AND:IE$^k$/MCC and 5c.c7:H57 receptor:ligand pairs, but were substantially reduced if not absent when 5c.c7 T-cells engaged the high affinity ligand IE$^k$/AEP or the lower affinity version IE$^k$/MCC. This may indicate that the interaction of AND TCR with its natural ligand can withstand higher molecular tension than the 5c.c7 TCR.

### TCR-imposed mechanical pulling forces are rare events

We next examined the frequency of TCR-imposed force events. By determining the proportion of force events above a calculated threshold (see Fig. 1c, left column and the Methods section), we found marked differences upon varying experimental conditions (see Fig. 2b). Notably, the results were consistent with above observations: The high-force fraction was largest for AND:IE$^k$/MCC and 5c.c7:H57 pairs (about 8–14% on gel-phase SLBs) and considerably reduced for the other experimental conditions (0–6% on gel-phase SLBs, 0–4% on fluid-phase activating SLBs). For T-cells engaging antigen on fluid-phase SLBs under scanning conditions, force events were registered between 2% (5c.c7:IE$^k$/MCC) and 6% (AND: IE$^k$/MCC) of all recorded smFRET events. Our analysis furthermore supported comprehensive comparison of different SLB compositions: The majority of high-force events was registered on activating gel-phase SLBs. A similar proportion could be seen under scanning conditions for AND:IE$^k$/MCC and 5c.c7:H57 pairs on gel-phase SLBs, but not for 5c.c7:IE$^k$/MCC and 5c.c7:IE$^k$/AEP. Forces were hardly detectable under activating conditions involving fluid phase SLBs, in line with a previous study of ours involving the use of T-cells engaging SLB-presented MFS[11]. Force events recorded under activating conditions preferentially occurred 10 min after cell seeding. In contrast, under scanning conditions force events became apparent within the first 10 min of cell seeding (Fig. 2c,

see Supplementary Table 2 for summary statistics). It should be noted that temporal analyses of this type are meaningful only to reveal trends in behavior due to the lack of synchronized T-cell antigen engagement with a number of incoming T-cells also settling during later stages of data acquisition.

Taken together, the large majority of MFSs did not experience forces. On gel-phase SLBs, the high-force fraction made up less than 14% of all measured single-molecule FRET events. This proportion was further reduced (less than 6%) on fluid-phase SLBs. We hence conclude that only a small fraction of the observed smFRET events experience a mechanical pull.

### Bound TCR ligands rarely experience mechanical forces

Next, we set out to determine molecular forces which are solely experienced by ligand-bound TCRs. Freely moving ligands embedded within fluid-phase SLBs are markedly reduced in lateral diffusion upon binding to TCRs within the immunological synapse[44,45]. We took advantage of this behavior to discriminate TCR-bound from free MFS-functionalized ligands. To this end, the smallest enclosing circle was computed for each single-molecule track. If its radius was below a calculated threshold (see the methods section for details), it was considered TCR-bound.

As shown in Fig. 3a and Supplementary Fig. 5, we observed the highest proportion of bound sensors upon use of stably binding H57-scF$_V$-functionalized MFS (about 75% in activating, more than 50% in scanning conditions). For the different TCR:pMHC pairs, 30–60% of MFS were engaging TCRs under activating conditions, compared to 20–30% under scanning conditions.

As expected, only immobilized but not mobile sensors were subjected to forces (Fig. 3b, see Supplementary Table 3 for summary statistics). In line with the findings of the previous section, we failed to detect forces under activating conditions, except for the AND TCR for which we noticed a small force-registering fraction. To visualize the number of sensors experiencing forces, we computed the differences of cumulative distribution functions (CDF) of synaptic sensors and sensors in the absence of cells (Supplementary Fig. 6). For all TCR:ligand pairs measured under activating conditions, CDF differences of bound and unbound sensors appeared visually similar and did indeed not exhibit a significant divergence (adapted KS-test, refer to the Methods section). Under scanning conditions, approximately 20% of bound AND:MFS-IE$^k$/MCC and 10% of bound 5c.c7:MFS-H57 events revealed forces (Fig. 3b) and CDF differences between mobile and immobilized events were distinct (Supplementary Fig. 6).

In summary, fluid-phase SLBs permitted discrimination between bound and unbound sensors. Under activating conditions, under which T-cells established tight immunological synapses, we failed to detect forces even when restricting the analysis to TCR-engaged sensors. Under scanning conditions, which promoted transient T-cell:SLB contacts, TCR-engaged sensors under force were detectable but were outnumbered more than 1:5 by force-free TCR-bound ligands (Fig. 3a, b).

### Global TCR:pMHC interaction lifetime is not affected by TCR-imposed mechanical forces

Finally, we assessed the impact of mechanical forces on the lifetime of synaptic TCR:pMHC interactions. While most protein:protein interactions become destabilized under force (termed slip bonds), stimulatory TCR:pMHC interactions have previously been described as catch bonds, which gain in duration when placed under force[26–28]. We quantified the average lifetime of TCR:pMHC interactions via single-molecule FRET microscopy using fluorescent H57-scF$_V$ labeling TCRs and fluorescent SLB-anchored IE$^k$/MCC or IE$^k$/AEP[15,46]. TCR:pMHC binding leads to measurable FRET by positioning acceptor and donor fluorophores in close proximity (Fig. 4a, b). The stability of TCR:H57-scF$_V$ bonds was confirmed by varying the incubation temperature

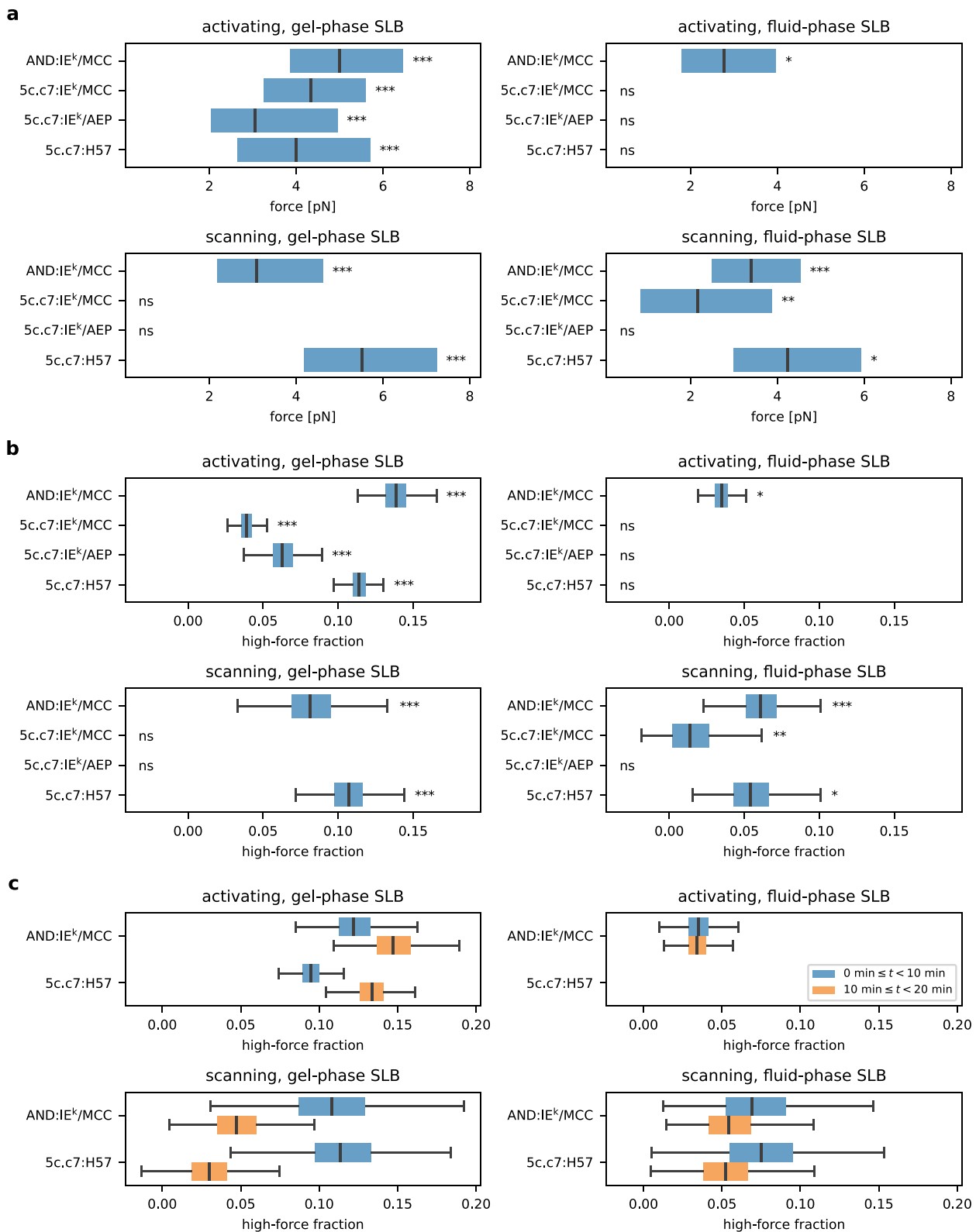

**Fig. 2 | TCR-imposed mechanical forces do not correlate with TCR:pMHC affinity. a** Single-molecule force quantification of investigated TCR:pMHC pairs. The number of recorded trajectories and mice are summarized in Supplementary Table 1. **b** Proportion plot for the recorded single-molecule TCR-imposed forces. All events with FRET efficiency below the calculated threshold (see Methods) are shown. The number of recorded trajectories and mice are summarized in Supplementary Table 1. **c** Temporal occurrence of the recorded force events in cell seeding intervals (first 10 min after seeding, and 10–20 min after seeding). The number of recorded trajectories and mice are summarized in Supplementary Table 2. Significant differences to the no-cell data are indicated by the asterisks in the corner of each plot. * – *p*-value < 0.05; ** – *p*-value < 0.01; *** – *p*-value < 0.001; ns – not significant. *P*-values were calculated employing one-sided KS tests adjusted for dependent datapoints within trajectories (see Methods for details). Boxes extend from the first to the third quartile, with the central line indicating the median. Whiskers ((**b**) and (**c**) only) extend to the most extreme data points within 1.5 times the interquartile range off the first and third quartiles. Source data are provided as a Source Data file.

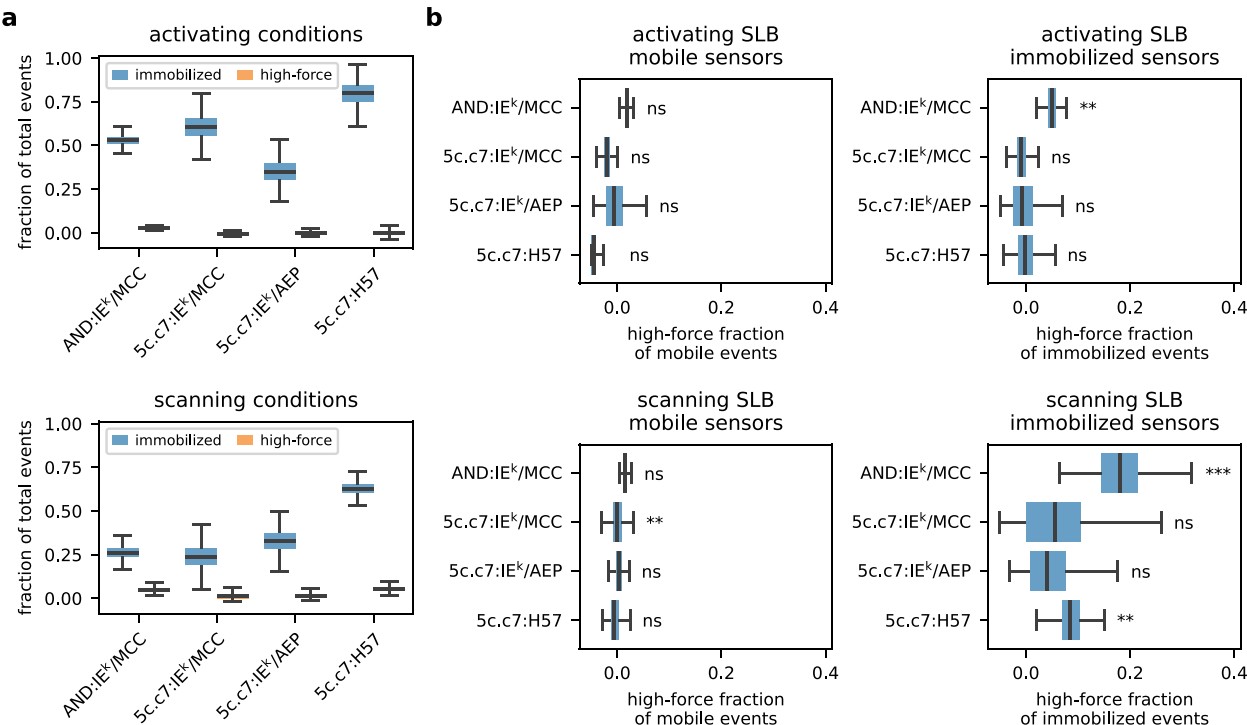

**Fig. 3 | TCR-imposed mechanical forces are rare events. a** Total amount of force events compared to total binding events. The number of recorded trajectories and mice are summarized in Supplementary Table 1. **b** Proportion of the molecular high-force events for the mobile and immobilized fraction of events. Significant differences to the no-cell data are indicated by the asterisks in the corner of each plot. * – *p*-value < 0.05; ** – *p*-value < 0.01; *** – *p*-value < 0.001; ns – not significant.

*P*-values were calculated employing one-sided KS tests adjusted for dependent datapoints within trajectories (see Methods for details). Boxes extend from the first to the third quartile, with the central line indicating the median. Whiskers extend to the most extreme data points within 1.5 times the interquartile range off the first and third quartiles. The number of recorded trajectories and mice are summarized in Supplementary Table 3. Source data are provided as a Source Data file.

(Supplementary Fig. 4a, b). The binding efficiency of H57-scF$_V$ to 5c.c7 and AND transgenic T-cells was tested via flow cytometry and a saturating amount of label was chosen (labeled and unlabeled moieties in a ratio of 1:5, Supplementary Fig. 4c). We measured synaptic TCR:pMHC lifetimes employing activating gel-phase SLBs, i.e. experimental settings giving rise to recordable TCR-imposed forces, and compared them with lifetimes determined using activating fluid phase SLBs, i.e. a regimen for which we were unable to detect forces. Of note, this approach is inadequate for measurements carried out under scanning conditions since ligand densities required to record sufficient data points within a feasible time frame are above the T-cell activation threshold[46].

Results are summarized in Fig. 4c and detailed statistics are shown in Supplementary Table 4. For the 5c.c7:IE$^k$/MCC pair, lifetimes were around 10–15 s at 23 °C and 5 s at 27 °C. Measured lifetimes of synaptic 5c.c7 TCR interactions with IE$^k$/AEP ranged from 100 to 200 s (note that for lifetimes this long, the experimental method is inherently imprecise[46]). The AND:IE$^k$/MCC bond lasted about 75 s at 23 °C and 25 s at 27 °C. The results for 5c.c7 and AND cells interacting with IE$^k$/MCC agree well with earlier findings from single-molecule tracking data[45]. This is particularly noteworthy as H57-Fab binding has been found to increase bond lifetimes by an order of magnitude in at least two studies[21,47]. For the TCR:pMHC pairs investigated here, this appears not to be the case, as afore-mentioned tracking study[45] did not involve any H57-derived proteins. However, an influence of the H57-scF$_V$ on the measured lifetimes in our experiments cannot be completely ruled out.

Importantly, lifetimes were not affected by force as the experimental use of fluid- or gel-phase SLBs, which afforded low and high force regimens, respectively, did not change the outcome. In line with this observation, depolymerization of the cortical actin cytoskeleton

through cytochalasin D, a measure we chose to eliminate mechanical TCR-imposed forces[11], had no noticeable influence on TCR:pMHC lifetime in synapses of T-cells seeded on protein-functionalized gel-phase SLBs (Fig. 4d).

In summary, we tested various conditions known to evoke different levels of force exertion for their influence on synaptic TCR:pMHC binding kinetics, and found no changes in the global TCR:pMHC bond lifetime, indicating that the rare mechanical force events we observed did not notably impact the overall TCR:pMHC bond lifetime distribution of activating T-cells.

## Discussion

T-cells are highly motile and traverse a multitude of tissues in search of their cognate antigen. It is therefore just a small step to assume that biophysical processes impact antigen scanning efficiency by pushing and pulling on the passing-by T-cells. So far, it has remained an open question whether T-cells exploit TCR-imposed forces for regulatory purposes, or whether they rather stabilize the micro-environment so that receptor-ligand interactions can proceed undisturbed.

In search for an answer, several tools for measuring molecular forces have been developed, which may be categorized according to (i) their reliability to afford single-molecule read-out, (ii) analog vs digital read-out, (iii) sensor size, and (iv) the application of external force versus observation of endogenous force. The advantages and limitations of the various methods to quantify TCR-imposed forces have been extensively discussed elsewhere[48–50]. Our sensor platform is advantageous with regard to all of these aspects, as it permits analog single-molecule readout of TCR-imposed forces while minimizing size-induced side effects within the immunological synapse. Low sensor densities and strict filtering of recorded signal ensured exclusive

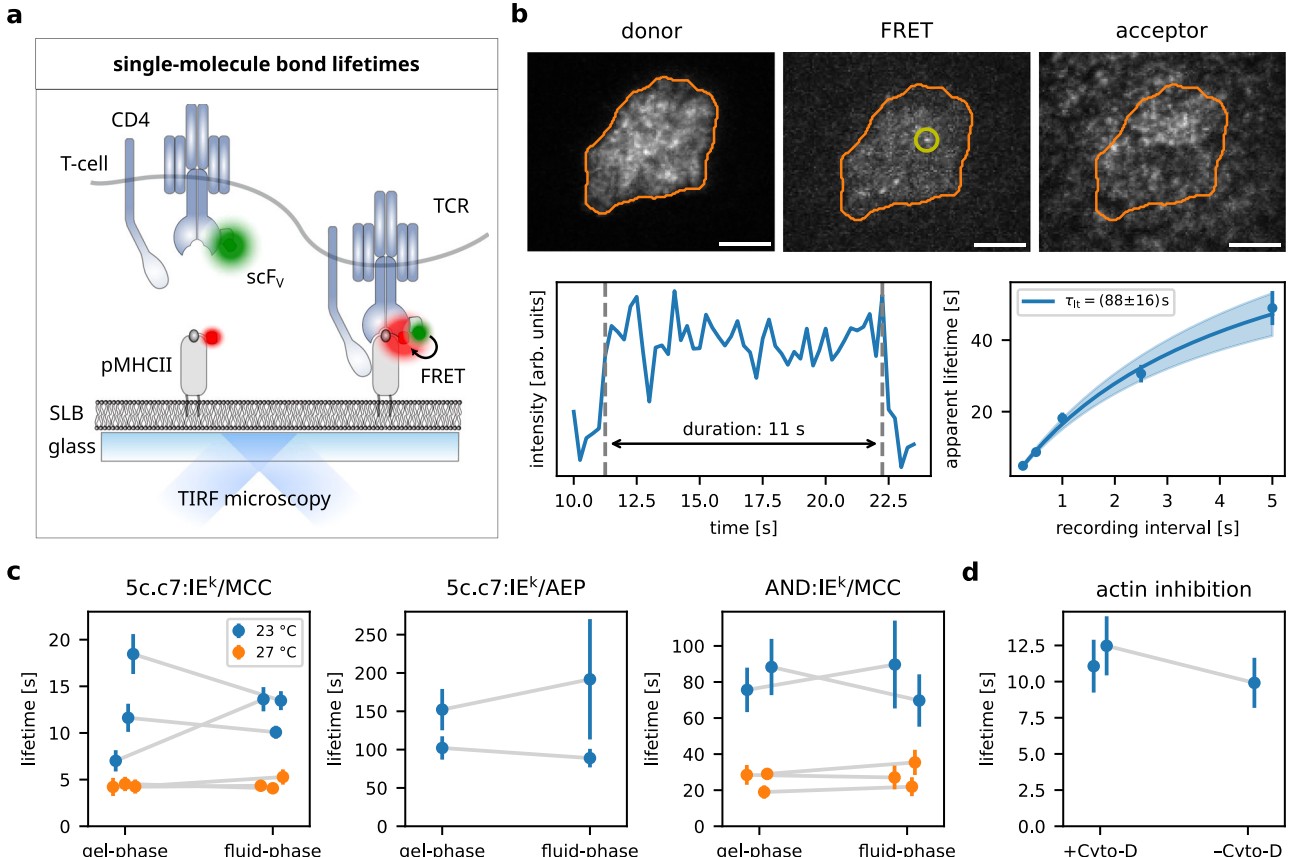

**Fig. 4 | Global TCR:pMHC interaction lifetime is not affected by TCR-imposed mechanical forces. a** Schematic representation of the quantification of single-molecule FRET events for the assessment of TCR:pMHC interaction lifetimes. Supported lipid bilayers (SLB) were decorated with $IE^k$/MCC fluorescently labeled at the C-terminus of the peptide (FRET acceptor). The TCR was labeled via fluorescent H57-$scF_V$ (FRET donor). Binding and unbinding could be tracked via the formation of single-molecule FRET signals. **b** Representative cell and intensity time trace of a single binding event and its subsequent lifetime analysis. Single-molecule FRET events were recorded for different time lags to estimate the effect of photobleaching and to eventually extract the average bond lifetime[46]. Scale bar: 5 µm. **c** Bond lifetime of indicated TCR:pMHC pairs at 23 °C and 27 °C on gel-phase and fluid SLBs. The results of the individual experimental days are plotted as paired data sets. Means and standard errors of the individual experiments are displayed as points and bars, respectively, and are summarized in Supplementary Table 4. **d** 5c.c7:$IE^k$/MCC bond-lifetime in the presence and absence of Cytochalasin D (inhibitor of actin polymerization) on gel-phase SLBs at 23 °C. Means and standard errors of the individual experiments are shown as points and bars, respectively, and are summarized in Supplementary Table 5. Source data are provided as a Source Data file.

analysis of single, isolated sensors. As a consequence, issues such as intra-molecular cross-talk, which have been found problematic with other methods[51], are circumvented by design. The spider silk peptide has a length of 2-3 nm in its coiled state and extends to about 8 nm at a tension of 10 pN. In its fully extended form this may constitute a non-negligible enlargement of the inter-membrane spacing within the immunological synapse (typically 13 nm[52]) and we cannot rule out secondary effects caused by the insertion of our force sensor. For instance, intrinsic force exertion may be affected as well as force directionality, in particular with respect to shear and bending (Fig. 1a). However, we observed a maximum of 7 pN force amplitude of the MFS-platform, which correlates to 2.2 nm extension. We can therefore assume that the partially extended sensor still fits into the narrow gap of the immunological synapse. SLBs also fail to emulate certain properties of APCs such as 3D structure, protein crowding, and elasticity. However, since current microscopy systems do not offer the required sensitivity for single-molecule force measurements in 3D, we believe that the presented approach offers the most direct and unbiased force quantification of surveilling T-cells.

With our sensor platform, we set out to determine the intrinsic behavior of T-cells while interfering as little as possible. In particular, we refrain from application of external forces onto T-cells and TCR:pMHC bonds[48]. We show that TCRs of CD4 + T-cells impose much lower mechanical forces on their ligands than expected. The majority

of the TCR-ligand pairs did not experience any force in the observation periods, a fact that has also been independently observed in a recent work using DNA hairpin-based sensors[51]: Forces only became detectable when arresting hairpins opened by tension, thus integrating the signal over extended time intervals. A small fraction of TCR:ligand bonds is subjected to minute forces (up to 5.5 pN for the 5c.c7 transgenic TCR, up to 6.5 pN for the AND transgenic TCR). This is considerably below the 15–20 pN range for which maximum discriminatory power has been suggested for highly stimulatory TCR:pMHC pairs[53]. While the exact receptor:ligand combinations employed here have not yet been characterized with respect to their response to external force application, the single-molecular sensitivity of 5c.c7 T-cells towards $IE^k$/MCC[2,54] suggests a similar force range. Notably, we observed differential force profiles employed by the investigated TCR:pMHC pairs, with AND:$IE^k$/MCC withstanding larger forces than 5c.c7:$IE^k$/MCC. This indicates differential force sensitivity for subsets of TCRs as previously suggested[53].

We observed the highest TCR-imposed forces for ligands immobilized on gel-phase SLBs, and much reduced forces for ligands attached to fluid-phase SLBs. Notably, recorded molecular forces exerted on engaged ligands on mobile platforms were effectively detectable only under conditions allowing antigen surveillance (scanning conditions), but not during immune synapse formation and maintenance (activating conditions). On gel-phase SLBs, forces are

likely generated by lateral transport of TCRs along the T-cell membrane[11], thus possibly leading to shear- and rotational stress on the receptor:ligand bond (analogous to the MFS-bound TCR ligand in Fig. 1a). Since the presence of shear forces has been implicated in elevated T-cell sensitivity[14,55], we expected this to be an additional factor influencing the TCR:pMHC interaction lifetime on gel- compared to fluid-phase SLBs. Direct measurement of the lifetime, however, showed no evidence of this. In fact, even inhibition of actin polymerization, which has been demonstrated to abrogate force generation[11,51], had no effect on measured lifetimes.

These findings suggest that inside the synapse forces are too rare and/or too weak to substantially influence the TCR:pMHC bond and in particular constitute evidence against catch- and slip bond formation as a fundamental mechanism underlying T-cell antigen recognition. Considering the necessity for T-cells to exactly probe the bond lifetime of their TCR:ligand pairs for ligand discrimination, it seems plausible to prevent mechanical pulling forces from disturbing the interaction in order to increase reproducibility. With sensors attached to fluid-phase membranes, which offer resistance only in perpendicular direction, synaptic forces appeared as extremely rare events detectable only for the high-affinity AND TCR (with ~5% of bound sensor data points). In contrast and depending on the TCR:ligand match, up to ~20% of binding events experienced perpendicular force exertion in synapses of scanning T-cells. This result is surprising in view that T-cells experience global pulling and pushing forces in the nanonewton regime[9,56]. This asymmetry suggests that T-cells employ efficient environment-stabilizing mechanisms in order to prevent excessive molecular forces at their TCR:ligand conjugates once the immunological synapse is established. These observations are in line with the hypothesis of force shielding elements within the immunological synapse and during early T-cell activation[34]. It is conceivable that the shielding is caused by the large number of adhesion factor interactions which share and buffer the globally experienced mechanical load[57], thereby efficiently reducing the tension on single TCR:ligand interactions. Hence, the molecular forces a TCR:pMHC bond is experiencing differ considerably depending if they occur in first-contact scenarios without further adhesion control, or if they proceed after immune synapse formation in a more stabilized biophysical environment.

The generation of forces within the immune synapse perpendicular to the T-cell membrane requires membrane fluctuations. The number of receptor:ligand pairs sharing the load due to local membrane displacement depends on the size of the displaced membrane patch, and consequently on membrane's bending rigidity. It has been shown that using magnetic tweezers to apply 10–20 pN to small membrane regions can lead to membrane deformations with radii of tens to hundreds of nanometers. Thus, for instance, a force that generates a membrane contortion of 100 nm radius would be spread across six molecules given a TCR density of about 200 molecules per $\mu m^2$ as found in in TCR microclusters[58]. In contrast, parallel forces, which are caused by TCR coupling to F-actin[51,55] are likely shared between receptor:ligand pairs to a lesser extent: The force would only be distributed among TCRs coupled to the same actin filament.

Based on the combined evidence presented here, we propose that T-cells primarily aim to prevent mechanical forces from disrupting molecular interactions by forming immune synapses. While it has long been speculated that T-cells could leverage mechanical forces to discriminate between ligands, this appears feasible, if at all, only during the initial receptor interactions at the microvilli tip as it penetrates the target cell's glycocalyx. Antigen recognition by and activation of T-cells may be a two-step process: Preliminary screening is performed via microvillus-located TCRs, in which the cells may benefit from force-enhanced sensitivity[50]. A positive outcome could subsequently lead to the formation of an extended contact zone in which TCR:pMHC interactions are shielded from forces for improved specificity.

However, direct observation for such a phenomenon is missing. In our study, we demonstrate that, following the initial contact of CD4 + T-cells, most TCR bonds experience minimal tension, and thus do not engage in or benefit from dynamic bonding.

## Methods

### Materials

HBSS (Hank's buffered saline solution, Merck KGaA, Germany); 1xPBS (Merck KGaA, Germany); FCS (fetal calf serum, Biowest, France); DPPC (1,2-dipalmitoyl-sn-glycero-3-phosphocholine; Avanti Polar Lipids, Inc., USA); POPC (1-palmitoyl-2-oleoylglycero-3-phosphocholine; Avanti Polar Lipids, Inc., USA); DGS-NTA(Ni) (1,2-dioleoyl-sn-glycero-3-[(N-(5-amino-1-carboxypentyl)iminodiacetic acid)succinyl] (nickel salt); Avanti Polar Lipids, Inc., USA); DMEM (Dulbeccos's Modified Eagle's Medium, Thermo Fisher Scientific, USA); BSA (Bovine serum albumin, Thermo Fisher Scientific, USA); T-cell medium: RPMI 1640 (Life technologies, USA) supplemented with 10% FCS (Biowest, France), 100 U/mL penicillin/streptomycin (Life technologies, USA), 2 mM L-glutamine (Life technologies, USA), 1 mM sodium pyruvate (Life technologies, USA), 0.1 mM non-essential amino acids (Lonza, Switzerland) and 50 $\mu$M β-mercaptoethanol (Life technologies, USA); Histopaque-1119 (Merck, USA); Alexa Fluor 555 C2 Maleimide (Thermo Fisher Scientific, USA); Alexa Fluor 647 C2 Maleimide (Thermo Fisher Scientific, USA); ICAM-1-10xHis (Sino Biological, China); B7-1-10xHis (Sino Biological, China); Guanidine hydrochloride (Merck KGaA, Germany); Tris base (Merck KGaA, Germany); NaCl (Merck KGaA, Germany); HRV 3 C protease (Pierce, Thermo Fisher Scientific, USA); Dibenzylcyclooctyne-maleimide (DBCO, Jena Bioscience, Germany); Tris(2-carboxyethyl)phosphin-hydrochlorid (TCEP, Merck KGaA, Germany).

### Animal model

5c.c7 αβ TCR-transgenic mice (Tg(Tcra5CC7,Tcrb5CC7)IWep, PMID: 1328464) bred onto the B10.A background were housed in groups of 2–5 per cage in the pathogen-free facility at the Medical University of Vienna, Austria. Mouse house conditions: 22 °C, 53% relative humidity, 12 h dark-light cycle. After cervical dislocation by trained mouse house personnel, spleens and lymph nodes were harvested from 12–16 weeks old gender-mixed mice.

Spleens of AND-TCR transgenic B10.BR animals (Tg(TcrAND) 53Hed, PMID: 2571940) were removed and sent in DMEM/1% BSA to the Medical University of Vienna on ice. The mice were genotyped by polymerase chain reaction or by cytometry and housed in groups of 2–5 animals per cage in the specific pathogen-free Core Facility Animal Models at the Biomedical Center of LMU Munich, Germany.

Mouse breeding and euthanasia were evaluated by the ethics committees of the Medical University of Vienna and approved by the Federal Ministry of Science, Research and Economy, BMWFW (BMWFW-66.009/0378-WF/V/3b/2016). All procedures to isolate lymphocytes and splenocytes from 8–12 weeks old gender-mixed mice were performed in accordance to Austrian law (Federal Ministry for Science and Research, Vienna, Austria), the guidelines of the Federation of Laboratory Animal Science Associations (FELASA), which match those of Animal Research Reporting In Vivo Experiments (ARRIVE), and the guidelines of the ethics committees of the Medical University of Vienna. Breeding and keeping of AND-TCR transgenic mice has been approved by the Government of Upper Bavaria, protocol 55.2-2532.Vet_02-21-4.

### Mouse T-cells

To obtain antigen-experienced 5c.c7 and AND murine T-cell blasts, $7.5 \times 10^6$/mL splenocytes were stimulated with 2 $\mu$M C18 reverse-phase high-performance liquid chromatography (HPLC)-purified MCC (88-103) peptide (sequence: ANERADLIAYLKQATK, Intavis, Germany) at 37 °C in an atmosphere of 5% $CO_2$ in T-cell medium composed of

RPMI 1640 (Gibco) supplemented with 10% FCS (Sigma), 100 U/ml penicillin/streptomycin (Gibco), 2 mM glutamate (Gibco), 1 mM sodium pyruvate (Gibco), 1× non-essential amino acids (Gibco) and 50 μM ß-Mercaptoethanol (Gibco). On day 2, doubling of the culture volume was accompanied by supplementing with 100 U/ml IL-2 (eBioscience). On days 3 and 5, cultures were split in a 1:1 ratio. Dead cells were removed by centrifugation on a cushion of Histopaque-1119 (Merck, USA) on day 6. T-cells were used for experiments on day 7–9 after initial stimulation. Note that this has been previously described[40].

## Protein expression and refolding
The TCR β-reactive H57 single-chain fragment (scF$_V$) (J0, GenBank: MH045460.1) and the fluorescently labeled H57 scF$_V$ (J1, GenBank: MH045461.1) were produced via expression in *E. coli* BL-21 as inclusion bodies employing the pET system (Novagen). They were subsequently refolded in vitro via dialysis in order to dilute the concentration of guanidine hydrochloride within the refolding buffer (50 mM Tris pH = 8.0, 200 mM NaCl, 1 mM EDTA). After concentration employing 10 kDa cutoff spin concentrators (Amicon) in 50 μM Tris-(2-carboxyethyl)phosphine hydrochloride (TCEP, Pierce), scF$_V$s were purified using Superdex 200 (GE Healthcare) size exclusion chromatography in PBS. The product was then concentrated to 1 mg/ml in 50 μM TCEP and permitted to react with a 3-fold excess of Alexa Fluor 555 C2 Maleimide (Thermo Fisher Scientific) at room temperature for 2 h. Surplus fluorophore was removed by gel filtration and proteins concentration was adjusted to 0.5 mg/ml in 1xPBS and 50% glycerol. Note that this has been previously described[39,58].

The murine MHC class II molecule IE$^k$ α subunits (with a 6x histidine-tag) and the β subunits (with a 6x histidine-tag) were expressed in E. coli as inclusion bodies and refolded in vitro with a placeholder peptide (ANERADLIAYL[ANP]QATK) for later exchange with fluorescently labeled peptides. Refolded IE$^k$/ANP was purified via nickel−nitrilotriacetic acid (Ni-NTA)-based affinity chromatography followed by S200 and S75 gel filtration. The peptides MCC (ANERADLIAYLKQATKGGSC), T102S (ANERADLIAYLKQASKGGSC), K99R (ANERADLIAYLRQATKGGSC) Null (ANERAELIAYLTQAAKGGSC) and AEP (ADGVAFLKAATKGGSC) were site-specifically labeled via Alexa Fluor 647 C2 Maleimide (Thermo Fisher Scientific, USA) and purified via preparative C18 reverse phase HPLC as described in the section "Molecular Force Platform Synthesis". After peptide exchange, fluorescently labeled IE$^k$/MCC derivatives were purified via S75 gel filtration and stored at −20 °C in 1 x PBS and 50% glycerol before use. Note that this has been previously described[39,59].

Murine recombinant ICAM-1-10xHis and B7-1-10xHis were purchased from Sino Biological (China).

## Molecular force platform synthesis
The MFS-IE$^k$/MCC and MFS-H57 constructs are the same batch as our previous publication[11], whereas the MFS-IE$^k$/AEP construct was newly synthesized and quality controlled according to the established protocols[11]. In principle, as anchor unit we used a his-tagged monovalent streptavidin which can bind the biotinylated spring unit. The force-sensing unit (spring unit) was covalently attached to the base of a recombinant site-specifically-modified IE$^k$/MCC moiety (functional unit). The purified molecular force sensor (spring unit coupled to the functional unit) was subsequently added to SLBs decorated with anchor units.

To produce the anchor unit (recombinant mSAv-3xHis6), biotin-binding deficient subunits ("dead", with a C-terminal His6-tag) and functional subunits ("alive", with C-terminal 3 C protease cleaving site followed by Glu6-tag) were expressed in E. coli (BL-21) using the streptavidin-pET21a expression plasmid (Novagen). Inclusion bodies were dissolved at 10 mg/mL in 6 M Guanidine hydrochloride, mixed at a ratio of 1 "alive" to 3 "dead" subunits and refolded in 100 volumes of

1xPBS. The salt concentration was reduced by a series of concentration via a 10 kDa cutoff spin concentrator (Amicon) and dilution with 20 mM TRIS pH 7.0. mSAv-3xHis6 was eluted from a MonoQ 5/50 column (Cytiva) at 20 mM TRIS pH 7.0, 240 mM NaCl. The Glu6-tag was removed via digestion overnight with human 3 C protease (Pierce). Size exclusion chromatography (Superdex 200, Cytiva) was used to yield the purified anchor unit. Note that this has been previously described[11,60].

Fluorescently labeled and unlabeled MFS peptides (spring units) with sequences: MFS peptide (biotin-N-GGCGS(GPGGA)5GGKYGGS-K(ε-N3)), MFS$_C$ peptide (biotin-N-GGGGS(GPGGA)5GGKYCGS-K(ε-N3)), and MFS$_O$ peptide (biotin-N-GGGGS (GPGGA)5 GGKYG GS-K(ε-N3)) were purified employing preparative C18 reverse phase HPLC (5 μm particle size, 250 × 21.2 mm, Agilent) with a linear gradient of 0.1% triflouracetic acid (TFA) in deionized water (buffer A) and 0.1% TFA, 9.9% deionized water, 90% acetonitrile (buffer B) at a flow rate of 5 ml/min over 80 min. For labeling the peptides, they were incubated first with Alexa Fluor 555 maleimide (Thermo Fisher Scientific) at 1.5x molar excess for 2 h at room temperature, then with Alexa Fluor 647 succinimidyl ester (Thermo Fisher Scientific) at 1.5x molar surplus and 330 mM NaHCO$_3$ for 2 h at room temperature. After each step, products were purified via HPLC and lyophilized. Note that this has been previously described[11].

TCRβ-reactive H57 antibody single chain fragments with a C-terminal unpaired cysteine residue (J4, H57 cys scF$_V$) were produced as described in the section "Protein expression and refolding" and subsequently labeled with dibenzylcyclooctyne-maleimide (DBCO, Jena Bioscience), were created as described in the section "Protein expression and refolding". IE$^k$ α featuring an additional C-terminal cysteine residue and IE$^k$ β were expressed as inclusion bodies and refolded in the presence of MCC 88-103 peptide (with the sequence H-ANERADLIAYLKQATK-OH) or a recently described high affinity derivate of MCC (AEP, with the sequence H-ADGVAFLKAATK-OH)[37] for 2 weeks. Correctly folded IE$^k$ molecules were isolated via an in-house built 14-4-4 antibody column, reduced with 25 μM TCEP followed by S200 gel filtration. Monomeric IE$^k$ MCC with unpaired cysteine residue was collected and coupled to DBCO. Note that this has been previously described[11].

MFS peptides were mixed with DBCO conjugated proteins and 25 Vol % 2 M TRIS pH 8.0 at a molar ratio of at least 5:1 and incubated for 2 h (room temperature). A monomeric avidin column (Pierce) was used to remove unreacted protein. After elution with biotin, unconjugated spring was separated from MFS conjugates using size exclusion chromatography (Superdex 75 or 200 column, GE Healthcare).

## Supported lipid bilayer formation
Gel-phase SLBs were created by using DPPC as carrier lipid, whereas fluid SLBs were created using POPC. Dissolved in chloroform, carrier lipids were mixed with 2 mol-% DGS-NTA(Ni) and subsequently dried under a nitrogen stream for 20 min. After resuspension in 1 mL 1xPBS, they were sonicated for 10 min in an ultrasound water bath (USC500TH, VWR, England) at room temperature (fluid SLBs) or 55 °C (gel-like SLBs). The resulting small unilamellar vesicle solution was diluted to 125 μM using 1xPBS.

The original cover slip of an eight-well chamber (Nunc Lab-Tek, Thermo Fisher Scientific, USA) was replaced by attaching a plasma-treated (10 min; PDC-002 Plasma Cleaner, Harrick Plasma, USA) microscopy cover slip (MENZEL-Gläser Deckgläser 24 × 60 mm #1.5) using duplicating silicone (Twinsil soft 18, picodent, Germany). The vesicle solution was incubated for 20 min at room temperature (fluid SLBs) or 55 °C (gel-like SLBs) to form lipid bilayers. For calcium flux ligand titration experiments, a 16-well chamber (Nunc Lab-Tek, Thermo Fisher Scientific, USA) was used instead. SLBs were subsequently washed with 1xPBS at room temperature to remove excess vesicles.

His-tagged proteins (for activating conditions: 10 ng mSAv-3xHis6, 30 ng ICAM-1, 50 ng B7-1; for scanning conditions: 0.4 ng mSAv-3xHis6, 30 ng ICAM-1) were added in 50 μL 1xPBS to each well and incubated for 60 to 75 min at room temperature. Afterwards, the bilayer was washed with 1xPBS. SLBs were finally incubated for 20 min with MFS variants (for activating conditions with MFS-H57: 10 ng MFS$_0$-H57 (resulting in 50–100 molecules/μm$^2$), 30 to 100 pg MFS-H57 for each experimental day adjusted to reach densities suitable for single-molecule experiments; for activating conditions with MFS-IE$^k$/MCC or MFS-IE$^k$/AEP: 15 to 20 ng MFS$_0$-IE$^k$/MCC (resulting in 50–100 molecules/μm$^2$), 30 to 100 pg MFS-IE$^k$/MCC or MFS-IE$^k$/AEP for each experimental day adjusted to reach densities suitable for single-molecule experiments; for scanning conditions: 30 to 100 pg MFS for each experimental day adjusted to reach densities suitable for single-molecule experiments) in 50 μL 1xPBS per well for binding to the mSAv-3xHis6. Bilayers were washed to remove excess MFS. Immediately before adding cells, the buffer was exchanged for HBSS containing 2% FCS.

For titration experiments, molecular densities were determined by dividing the fluorescence signal per pixel by the single-molecule brightness recorded at the same settings, considering the effective pixel width of 160 nm. In single-molecule samples, densities were determined by counting the number of molecules and dividing by the area of the field of view.

## Single-molecule microscopy setup

For live cell imaging with single-molecule resolution, we used objective-type total internal reflection (TIR) illumination of fluorophores which was achieved using an objective with high numerical aperture (α Plan-FLUAR 100x/1.45 oil, Zeiss, Germany). Fluorophores were excited with 640 nm (OBIS 640, Coherent, USA) or 532 nm (LCX-532L with L1C-AOM, Oxxius, France) laser light coupled into an epifluorescence microscopy (Zeiss, Germany). The emission beam was separated from the excitation light via a quad-band dichroic mirror (Di01-R405/488/532/635-25×36, Semrock Inc., USA).

Using a beam splitter device (Optosplit II, Cairn Research, UK) comprising a dichroic mirror (FF640-FDi01-25×36, Semrock Inc., USA) and bandpass filters (ET570/60 m and ET675/50 m, Chroma Technology Corp, USA), the fluorescence emission path was split and projected side-by-side onto the chip of an electron multiplying charge-coupled device (EM-CCD) camera (Andor iXon Ultra 897, Andor Technology Ltd, UK). The microscope and peripherals were controlled by using the SDT-control software developed in-house.

In addition, we used a 405 nm laser (iBeam Smart 405-S, Toptica Photonics AG, Germany) to image Fura-2-loaded cells or reflection interference contrast microscopy (IRM) to visualize the cell contours. For the latter, white light emitted by a mercury arc lamp (HBO 100, Zeiss, Germany) was coupled into the emission beam path via a dichroic mirror (DMSP680B, Thorlabs, Inc., USA) and enabled/disabled via an electronic shutter (VS14, Vincent Associates, USA).

## Single-molecule FRET measurement for molecular force quantification

T-cells were seeded onto sensor-functionalized SLBs. Once cells started spreading (typically after 2 to 3 min), image sequences were recorded by alternately exciting the donor fluorophore using the 532 nm laser and the acceptor using the 640 nm laser. For a typical video, this process was repeated 300 times with illumination durations of 5 ms and pauses of 5–85 ms between frames to allow for camera read-out and to adjust the recording rate. In addition, cell outlines were recorded at the beginning and end of each video either using 405 nm excitation (if cells were loaded with Fura-2) or by means of IRM.

For flatfield correction, SLBs with high densities of MFS (5–100/μm$^2$) were employed. First, an image upon 640 nm laser excitation was recorded, followed by complete photobleaching of the acceptor fluorophores at high laser intensity. Finally, a frame upon 532 nm excitation was recorded.

For image registration, images of fluorescent beads (TetraSpeck, Invitrogen, USA) were recorded upon 523 nm laser excitation. Note that this has been previously described[11]. A detailed protocol has also been published[36].

## Conversion between FRET efficiency and force

The sensor has been calibrated as previously described[11]. In summary, Brenner et al.[61] found a linear relation between force $F$ and dye separation $r$, $F(r) = \frac{r - b \cdot n - c}{a \cdot n}$ with $a = 0.0122$ nm pN$^{-1}$ and $b = 0.044$ nm. For our sensor ($n = 29$ amino acids), we found $c = 2.4$ nm. Inversion of the relation between FRET Efficiency $E$ and $r$ for Förster radius $R_0 = 5.1$ nm, $E(r) = \left(1 + \left(\frac{r}{R_0}\right)^6\right)^{-1}$, yields the desired map $F(E)$.

## Single-molecule force data analysis

Our publicly available software[62] was used to perform the following tasks: loading of raw data, registration of donor and acceptor emission channels, localization of fluorescent signals, single-molecule tracking, single-molecule intensity determination, nearest-neighbor analysis to discard overlapping signals, flatfield correction, computation of apparent FRET efficiencies and stoichiometries, stepwise bleaching analysis to remove signals from multiple emitters, computation and application of FRET correction factors (bleed-through, direct acceptor excitation, detection and excitation efficiency factors), filtering based on FRET efficiency and stoichiometry, segmentation of images of cells for cell outline determination, saving of resulting single-molecule data. Note that this has been previously described[11]. A detailed protocol has also been published[42].

## Database management and evaluation criteria

Experimental data was collected over extended time frames. To guarantee reproducibility and consistency a tight control frame work was established which was adhered to during each experimental day. The collected data from the smForce experiment and the simultaneously measured calcium response data was organized on a file server. After completed quantification of the molecular force and calcium flux response, each data set was evaluated to be part of the integrated result database. Evaluation criteria were the functional state of the T-cell population (positive vs negative control), the functional state of the SLBs (activating vs scanning conditions), the mobility of the mobile SLBs, the absence of force events within the no-cell data, and the response of the positive control of the force experiment (MFS-H57 on gel-phase SLBs). In case of the extraction of immobilized signals from the mobile fraction, the absence of immobilized signals within the no-cell data, was included as another evaluation criterium. Consistently treated data was then pooled to arrive at the final quantification of the molecular force for each investigated condition.

## Determination of the uncertainty of measurement

To calculate the uncertainty, FRET efficiency values from a medium-sized, representative dataset (IE$^k$/MCC-functionalized MFS, activating conditions, fluid SLB) without cells were resampled 1000 times while preserving single-molecule tracks. From each sample, mean and standard deviation were calculated as maximum likelihood estimates of normal distributions' μ and σ parameters. Mean and standard deviation of these estimates serve as parameter value and standard error estimates.

100 force values $F_i$ between 0 and 9 pN were mapped to their corresponding FRET efficiency values $E_i$. For each $F_i$, its uncertainty estimate $\Delta F_i$ was computed according to $\Delta F_i = F(E_i - \Delta E) - F_i$, where $\Delta E$ is the standard error of µ as described above and $F(E_i - \Delta E)$ is the force corresponding to the FRET efficiency $E_i - \Delta E$. The maximum $\Delta F_i$ was subsequently computed.

### Hypothesis tests with single-molecule force data

Datapoints from the same single-molecule track may be correlated, precluding the use of regular statistical tests such as the KS test. Instead, we employed permutation tests and ensured that tracks were kept intact during resampling[43]. As null hypothesis we defined that FRET efficiency distributions originating from sensors on SLBs without cells are indistinguishable from sensors within the interaction area between cells and SLBs, as alternative hypothesis we stated that TCR-engaged sensors experience greater force and therefore reduced efficiency. Hypotheses were tested using the maximum difference of CDFs as a test statistic in the permutation tests (9999 resamples), thereby effectively performing a one-sided KS test on the tracking data (Figs. 2 and 3, Supplementary Tables 1–3). The difference between distributions of mobile vs. immobilized sensors (Supplementary Fig. 6) was determined likewise.

### Analysis of the high-force (= low-efficiency) fraction

In order to determine the proportion of single-molecule FRET events originating from sensors subjected to force, first a threshold was computed: From data recorded without cells under otherwise same conditions (i.e., SLBs with the same lipid and protein composition), the $f = 0.05$ quantile $q_f$ was determined as a threshold with a false-positive rate of $f = 5\%$. Using pooled cell-derived single-molecule data, the number of events $n_{low}$ with a FRET efficiency below $q_f$ was divided by the total number of events $n_{total}$, yielding the proportion of force-affected sensors with an expected false-positive rate of $f$. Therefore, the actual high-force fraction was calculated according to $n_{low}/n_{total} - f$. Note that variation of the $f$ had no substantial influence on the results. In order find an estimate for distribution of high-force fractions, the calculation was repeated 1000 times with randomly resampled data. Resampling was performed such that individual single-molecule tracks were kept intact as they may contain correlated data. The resulting distributions are displayed in Figs. 2b, c and 3b as box-and-whiskers plots. Boxes extend from first to third quartile with the median depicted as a solid line. Whiskers extend to the most extreme data points within 1.5 times the interquartile range off the first and third quartiles. Means and standard deviations of the distributions are given in Supplementary Tables 1–3 as high-force fraction proportions and standard errors, respectively.

To quantify the magnitude of forces experienced by sensors, the distribution of forces corresponding to the high-force fraction was isolated as follows: Average shifted FRET efficiency histograms (range: −0.5 to 1.5, number of shifts: 40, number of bins determined by Sturges's rule, i.e., $1 + \lceil \log_2(n) \rceil$) yielding graphs (i.e., pairs of $x, y$ coordinates) of the probability density functions (PDFs) of cell-free as well as cell-impacted FRET efficiency data. We subsequently introduced two scaling parameters $a, b$ for the cell-free PDF. $a$ was used to scale the $x$ coordinate according $0.87 + a(x - 0.87)$, effectively scaling the peak width around the FRET efficiency of 0.87 (which corresponds to zero force). $b$ scaled the peak height via $by$. Non-linear least squares minimization (scipy.optimize.least_squares function[63]) was used to find the best approximation of the scaled cell-free PDF to the cell-impacted PDF in the low-force regime (FRET efficiency > 0.75). The thusly scaled cell-free PDF was then subtracted from the cell-impacted PDF, yielding an estimate of the FRET efficiency PDF of the high-force (= low efficiency) fraction. Using the sensor calibration function, this was converted to a force PDF. Since a PDF has to be strictly non-negative, all negative y values were

replaced with 0. Finally, the quartiles as shown in Fig. 2 and Supplementary Table 1 were computed.

### Analysis of the mobile fraction

For the distinction between mobile and immobilized MFS tracks, the smallest enclosing circle of the track was computed using the spatial.smallest_enclosing_circle function from the sdt-python Python package[64]. Its reduced radius was computed by dividing the radius by the square root of the duration of the track, since the squared radius is expected to grow linearly with time assuming free diffusion. Any track with a reduced radius of less than 0.35 µm/s$^{0.5}$ was considered immobilized. To ensure consistency, all tracks containing less than five observations were discarded. Note that this has been previously described[11]. In order find an estimate for distribution of mobile fractions, the calculation was repeated 1000 times with randomly resampled data. Resampling was performed such that individual single-molecule tracks were kept intact as they may contain correlated data. The resulting distributions are displayed in Fig. 3a as box-and-whiskers plots. Boxes extend from first to third quartile with the median depicted as a solid line. Whiskers extend to the most extreme data points within 1.5 times the interquartile range off the first and third quartiles.

### Single-molecule FRET measurement for quantification of interaction lifetimes

Approximately 20% of TCRs on T-cells were labeled using AF555-H57-scF$_V$ in order to reduce bleed-through into the FRET acceptor channel (1 part labeled H57-scF$_V$ was pre-mixed with 4 parts unlabeled H57-scF$_V$ to reach a final labeling mass of 120 ng per 1 Mio T-cells). Labeling was performed on ice for 20 min, excess scF$_V$ was removed by washing with cold imaging buffer (HBSS + 2% FCS). Cells were subsequently kept on ice. The buffer in SLB-containing Labtek wells was exchanged with imaging buffer, and $10^5$ T-cells were seeded immediately before measurement.

smFRET events were recorded using 10 ms illumination time. Acceptor fluorophores were excited using a 640 nm laser at the beginning and the end of the recording, whereas donor fluorophores were excited distinct number of times using varying delays. Lifetimes were recorded at 23 °C and 27 °C for various receptor:ligand pairs.

### Lifetime data analysis

We used our publicly available software[65] to load the data, perform image registration for donor and acceptor emission channel, correct acceptor emission images for donor bleed-through, localize and track single-molecule signals, apply changepoint detection to ensure single-step appearance and disappearance of analyzed FRET events, and manually verify each single-molecule trajectory. Track lengths were subjected to survival analysis to determine the apparent lifetime for each recording interval. Fitting apparent lifetime vs. recording interval curves with the appropriate model allowed for disentanglement of unbinding and photobleaching contributions, yielding the characteristic bond lifetime. Note that this has been previously described[46].

### Calcium flux measurements

The ratiometric dye Fura-2-AM[66] (Life technologies, USA) was used as an intracellular reporter on calcium levels. $5 \times 10^5$ T-cells were incubated with 2 µM Fura-2-AM in T-cell medium for 15 min at room temperature and washed with imaging buffer (HBSS with 2% FCS).

For antigen titration experiments, calcium response was recorded at room temperature, as well as 37 °C. Temperature control was carried out with a heating unit and a box enclosing the microscope. $1 \times 10^5$ T-cells were seeded onto SLBs decorated with increasing densities of IE$^k$/MCC (agonist), IE$^k$/T102S (weak agonist) or IE$^k$/Null (non-agonist), as well as unlabeled B7-1 and ICAM-1 (100 µm-2). Up to 12 different conditions were recorded simultaneously using 16-well Lab-Tek

Chambers (Nunc). A calcium multiplex setup was used to excite Fura-2-AM using a monochromatic light source (Leica) coupled into an inverted Leica DMI4000B microscope which was equipped with a 20x objective (HCX PL Fluotar 20x, NA = 0.5, Leica), the dichroic beamsplitter FU2 (Leica) and the bandpass filter ET525/36 (Leica). Excitation was continuously switched between 340 nm and 387 nm, which was achieved with the use of a fast excitation filter wheel (Leica) containing the excitation bandpass filters 340/26 and 387/11 (Leica). Images were recorded every 15 s for a total recording time of 20 min. An automated XY stage (Leica) allowed fast changes between different positions (i.e. wells).

T-cell quality was monitored in parallel to each single-molecule FRET experiment for each experimental day. T-cells were seeded onto functionalized SLBs decorated with either unlabeled B7-1 and ICAM-1 ($100\,\mu m^{-2}$) as negative control, or additionally unlabeled $IE^k$/MCC ($100\,\mu m^{-2}$) as positive control. Each SLB used for smForce experiments was controlled for its ability to elicit calcium response of seeded T-cells (i.e. activating and scanning conditions) for each experimental day. The ratiometric dye Fura-2 was excited with a monochromatic light source (Polychrome V, TILL Photonics, Germany) coupled to a Zeiss Axiovert 200 M, equipped with a 10× objective (UPlanFL N 10x, NA = 0.3, Olympus, Japan), a long-pass filter (T400lp, Chroma Technology, USA), an emission filter ET510/80 (Chroma Technology, USA), a 1.6× tube lens, and an EM-CCD camera (Andor iXon 897, Andor, UK). The dye was excited at 340 and 380 nm with illumination times of 75 and 30 ms for a total recording time of 15 min with a rate of 1 image per second. Experiments were carried out at room temperature.

Calcium image analysis was performed using our custom-made MATLAB (Mathworks, Inc., USA) software described in ref. 11.

### Flow cytometry

For cell surface labeling, $5 \times 10^5$ T-cells were labeled with AF555-H57-scF$_V$ (250 µg/mL) for 15 min on ice, and washed 2 times in FACS buffer (1x PBS, 1% BSA, 0.02% NaN3). For titration experiments, the following mass of the AF555-H57-scF$_V$ was used: 20, 200, 600, 1800, 5400, and 16200 ng per 1 Mio cells. For time curve experiments 1800 ng per 1 Mio T-cells was used. After washing T-cells were placed on ice, room temperature or into a 37 °C water bath. Samples were removed from the respective temperatures after the following intervals (0, 5, 10, 30, 60, and 120 min) and immediately analysed. Unlabeled cells were used as a reference and measured as separate sample. Samples were analyzed on the Cytek Aurora (Cytek Biosciences). Data derived from flow cytometry measurements were analyzed with the FlowJo v10 software (BD Biosciences). The lymphocytes were gated using FSC vs SSC. The singlets were gated based on the area vs height of FSC.

### Statistics and reproducibility

No statistical method was used to predetermine sample size. We excluded datasets from the analysis if any of the critical controls described in the section "Database management and evaluation criteria" failed. Accepted datasets were filtered as outlined in the sections "Single-Molecule Force Data Analysis" and "Lifetime Data Analysis". Allocating samples into experimental groups was not applicable. Blinding was therefore not relevant to this study.

### Reporting summary

Further information on research design is available in the Nature Portfolio Reporting Summary linked to this article.

### Data availability

The data generated in this study have been deposited in the TU Wien Research Data database under accession code dccvp-w7q74 (URL: https://doi.org/10.48436/dccvp-w7q74)[67]. Source data are provided with this paper.

### Code availability

We provide the Python code for single-molecule FRET analysis[62] (https://github.com/schuetzgroup/fret-analysis) as well as the underlying Python library[64] (https://github.com/schuetzgroup/sdt-python). Additionally, we provide the Python code for single-molecule Lifetime analysis[65] (https://github.com/schuetzgroup/smfret-bondtime). Code to produce the figures and tables for this article has been deposited alongside the data at https://doi.org/10.48436/dccvp-w7q74[67].

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

## Acknowledgements

This research was funded by: Austrian Science Fund (FWF) project 10.55776/P32307-B (L.S., J.G., G.J.S.). Austrian Science Fund (FWF) project 10.55776/P30214-N36 (L.S., G.J.S.). Austrian Science Fund (FWF) project 10.55776/P25775-B2 (F.K., J.B.H.). Austrian Science Fund (FWF) project https://doi.org/10.55776/PAT3728424 (J.G.). German Research Council (DFG) project OB 150/7-1 (J.W., R.O.). Boehringer Ingelheim Fonds (RP). Vienna Science and Technology Fund (WWTF) LS13-030 (J.G., F.K., L.S., J.B.H., G.J.S.).

## Author contributions

conceptualization: J.G., G.J.S., and L.S.; data curation: J.G. and L.S.; formal analysis: J.G. and L.S.; funding acquisition: J.G., J.B.H., and G.J.S.; investigation: J.G. and L.S.; methodology: F.K., J.G., J.W., L.S., P.H., R.P., R.O., V.M.; project administration: J.G., G.J.S., and J.B.H.; resources: F.K., J.G., J.W., R.O., R.P., P.H., and V.M.; software: L.S., and J.G.; supervision: J.G., G.J.S., and J.B.H.; validation: J.G. and L.S.; visualization: L.S., and J.G.; writing—original draft preparation: J.G. and L.S.; writing—review and editing: G.J.S., J.G., J.B.H., J.L.T.H., and L.S. All authors have read and agreed to the published version of the manuscript.

## Competing interests

The authors declare no competing interests.
