## [Peer Review File · Nature Communications]

CD4+ T-cells create a stable mechanical environment for force-sensitive TCR:pMHC interactions

Corresponding Author: Dr Janett Goehring

Version 0:

Reviewer comments:

Reviewer #1

(Remarks to the Author)

Manuscript Summary: Schrangl et al., present a manuscript which builds upon recent publications leveraging a spider silk protein-derived tension gauge tether system to measure forces in concert with FRET and TIRF implementations. This system is elegantly designed to take advantage of supported lipid bilayers (SLBs) in presenting peptide-MHC (pMHC) to T cells in order to measure forces exerted on pMHC through TCR engagement. The authors conclude from these efforts that once the immunologic synapse has formed, there appear to be few force enhanced events, and that those that do occur are of significantly lower forces, approximately 4-7 pN or less, than those found to activate catch bonds, structural transitions in TCRs and optimal activation of T cells, 10-15 pN.

Review: The broad conclusions of the paper if taken at face value are at odds with current interpretation of experimental evidence from several active mechanobiology-oriented T cell laboratories. Pointedly, this runs contrary to work presented in Hu et al., 2024 and prior work from that group. Thus, the questions raised are significant and should be taken seriously. This reviewer can find no fault in the implementation of the several complex technologies necessary to complete this work. We also agree with the authors that the question posed herein is a difficult one to answer unambiguously, given the challenge of measurement without perturbing the system such that the result becomes biased beyond use. We also note that the authors have provided qualifying statements concerning the use of SLBs vs APCs and we find their statements sufficient with some exceptions (see below). With this in mind, we ask that the authors provide clarification or additional discussion of the following points prior to consideration of publication.

1. The TCR-pMHC systems utilized are characterized with traditional biochemistry/biophysics, and 3D and 2D affinities are referenced, but it is not described what the optimal force regime of these systems are. Are they force responsive? Is that information available? We could not find such data in the referenced publications. This is important as the force responsiveness of TCRs is highly variable (See Liu et al., 2014, Akitsu et al, Science Advances 2024). There are many functional TCRs which operate without significant force enhancement, with suggestions that ligand density can substitute for TCR quality and these TCRs may belong to that subset. We have no expectation that such work should be required for publication, but an acknowledgment of this possibility is necessary.
2. The biophysical characterization of the tension-gauge tether system used here was thorough in the original Brenner et al., Nano Letters, 2016 publication and further elaborated and validated technically in the prior publications from this laboratory. However, the authors made statements that the length of the tether is not problematic in face of the 13 nm spacing of the immunologic synapse that are difficult to evaluate. If, in order to attain a 10 pN force, which according to Brenner, the tether elongates to 8 nm (or to 6 nm at 5 pN) according to Fig 3b therein, this does not leave much room for the pMHC or TCR proper without significant angular changes. For the pMHC with the tether inserted, we suspect that the degrees of freedom are sufficient to allow this, but given the TCR holocomplex with attendant CD3 subunits, such angular distortions may be disallowed, leading to potential torsional strain on the TCR-pMHC bond. This is, of course, speculative, but one cannot dismiss this or other unintended consequences of insertion of this rod-like, folded structure (via Brenner) out of hand. Please address this with further qualification in the discussion and if possible provide a diagrammatic illustration of such geometries either within or alongside Fig 1A-B. If the authors were able to show complete loss of the FRET fluorescence in this system, this would go a long way to convincing the reader that such geometry is attainable.
3. The authors call out the disadvantages of the DNA-tension-gauge tether system by virtue of not just its size, but also due to its digital character. The reviewer wonders whether this digital character may be an advantage in the TCR-pMHC system. One unintended consequence of the analog system is that it will diffuse the force (by ~ 0.35 pN/nm according to Brenner et

al.) as it is stretched, altering the force application from the T cell, presumably originating with the actomyosin machinery (Ma et al 2008, Feng et al., PNAS 2017). The DNA tethers, conversely, will transmit the force until a threshold is reached, at which point the tether unfolds. This behavior is analogous to the unfolding of the TCR under force and perhaps more compatible with a system tolerized to such digital behavior. Thus, the tether adds compliance to the system beyond any compliance of the membrane itself precisely during the ramp to "optimal" forces. This may affect the generation process within the cell. Could the authors please address this point?

4. Could the authors provide an direct statement of the error of measurement in the text for the FRET-distance values as well as the calculated force estimates? This can be from a representative experiment herein or from a prior publication. This would be helpful to the more general reader who will not want to dig through the supplementary material for such a rule of thumb when evaluating the data.

5. The single molecule FRET-based bond lifetime experiments are executed using scFV H57 bound to the TCR of interest. The lifetimes of TCR (and preTCR) bonds under force have been shown to be approximately 10-fold greater with H57 Fab addition (Das et al., 2015, Das et al., J Biol Chem 2016). This represents a potential modulation of the result as a consequence of the measurement system. This should be mentioned in the text.

6. We are unaware of documentation of direct coupling of pMHC in the membrane of APCs to cytoskeletal elements. There have been studies that have shown to motional inhibition modulated with the tail region (e.g. Edidin et al., PNAS 1994) as well as active transport of MHC proteins (e.g. Boes et al., Nature 2002). Could the authors provide relevant citations for the utilized viscosity models of the SLBs in the text? If possible, please address this directly in the introduction or early results section. Additionally, what is the potential for contribution of additional forces prior to and with MHC clustering on the APC?

In summary, the authors have presented much of their findings as a negative result. Yet, from this reviewer's perspective, the findings on these TCRs may be confirmatory of more recent work to characterize subsets of analog or digitally sensitive TCRs with heretofore uncharacterized single TCR force profiles on T cells engaged with multiple pMHC ligands during activation. Indeed, the experiments provided evidence for differential forces in a TCR-pMHC dependent manner, though this point is not emphasized. The experimental system provides optimal measurement in a lower force regime, 0-10 pN, than is optimal for high-performing TCR (10-25 pN), and places additional geometric constraints on the binding event. It may be that the confines of the intercellular space within the immunologic synapse limits the force profiles of these bonds and this work bolsters that argument. We accept that there is valuable information herein that will be useful to the research communities of T cell immunology as well as mechanobiology and thus would recommend publication only if the above questions are satisfactorily addressed. Additional data addressing any question would of course be welcome, but by further explanation within the text and more contextual framing of conclusions thereby, such concerns would likely be resolved.

(Remarks on code availability)

Reviewer #2

(Remarks to the Author)

The manuscript "CD4+ T-cells create a stable mechanical environment for force sensitive TCR:pMHC interactions" by Schragl et al. presents an in vitro study at the single molecule scale for assessing the occurrence of pulling force events and its influence on TCR-pMHC lifetime at the lymphocyte-surface interface.

This experimental configuration has been highly exploited in the last two decades, but the topic is of interest since the role and effect of force in TCR:pMHC interaction and function is currently debated in the literature, as presented in the introduction.

The study relies on a previously published molecular force sensor (MFS) based on FRET signal between acceptor and donors connected by an entropic peptidic spring, which has the advantage, in principle, of providing an analog force measurement at the single bond level with reduced disturbance on the system. The study exploits the systematic comparison between high and low ligand densities, as well as ligand mobility or not, and different TCR-pMHC pairs known to exhibit different binding properties. All these parameters are known to play an important role in the cell activation.

While the study seems overall solid, the current manuscript suffers from missing informations as well as confusing data presentation. Additionally, the discussion need to be reassessed in depth before further consideration for publication.

Major concerns:

1. the characterization of the presenting surface needs to be improved:

- the density of the ligands in the scanning vs activating condition is not measured
- how does it rely with densities used in figure S1F ?
- is Fig S1F obtained with mobile or immobile ligands ?
- following the previous questions, the terms 'scanning' and 'activating' are not properly justified and can be misleading
- for example, the authors could comment on the area of the contact zone in the two cases

2. the extraction of force and their distribution is incomplete and confusing:

- what is the relation between FRET efficiency and force ?
- how are the histograms in Fig S2 normalized ? why the integral of the difference in Fig S4 is not equal to zero ?
- the probability densities (Fig S2) and the difference with the control of out of cells (Fig S4) are presented in two separate figures and correspond to two different sections in the results. However the same data are concerned.
- in Fig S4 the densities are fitted before subtraction, whereas in Fig S6 they are not. It would be preferable to also avoid the fitting in S4.

- S4 and S6 also differ for the apparent choice of the y axis, which is not specified.

3. the authors claim in the discussion that one strength of their MFS is the analog force measurement. However, they introduce the "high force fraction", from which the observations they extract seem to be qualitatively redundant with the ones obtained from the force distributions.

This is especially the case since the statistical tests for the force distribution are performed with respect to zero force only and not between conditions (Fig 2A)

Also, they don't quantify what is "high-force". The presentation would gain in being more consistent throughout the manuscript.

4. Lifetime measurements:

- it would be good to superimpose the known data from the literature to the present measurements.

- the temperature has an obvious effect on the lifetime. But how did the authors assess its impact on cell activation/force exertion in their experiments ?

- the authors compare gel vs fluid phase, implicitly assuming that the vertical and lateral component of the force should affect the lifetime equivalently. They should explicit and justify better this assumption.

5. Discussion:

- while the title proposes an appealing hypothesis, it is not clear why the discrimination should be relevant in the so called 'activating' setting ?

- differential force sharing as a function of density should be better discussed: is it not expected to have less force exerted per bond if it is shared between more bonds ? would it be also useful to consider the total force exerted by the cell for a fair comparison ?

- are the cells in the same activation state in scanning and activating settings ?

Minor points:

1. Fig S1: how many cells ? define the error bars

2. Fig S3 is not called between S2 and S4.

3. Vocabulary: TCR-pMHC interaction under "strain" is used throughout while it seems to me that "stress" would be the correct word

4. discussion page 7: "inhibition of actin polymerization abrogates force generation": where are these data ?

5. Presentation of Fig 4C, D: i would include zero in the why axis to better emphasize the lack of difference between gel and fluid phase.

(Remarks on code availability)

Version 1:

Reviewer comments:

Reviewer #1

(Remarks to the Author)

The authors have clarified their manuscript considerably and have provided a study that raises many questions as well as provides significant insights. My only requirement for publication is that to assist in interpreting the findings for readers, the authors should specify within the Abstract that the measurements utilized pMHC anchored to supported lipid bilayers.

(Remarks on code availability)

Reviewer #2

(Remarks to the Author)

The authors have satisfactorily addressed my comments.

(Remarks on code availability)

We thank the reviewers for their insightful comments, which have led to a greatly improved version of the manuscript. Below we provide a detailed, point-by-point response to all suggestions and concerns raised.

Reviewer 1

1. The TCR-pMHC systems utilized are characterized with traditional biochemistry/biophysics, and 3D and 2D affinities are referenced, but it is not described what the optimal force regime of these systems are. Are they force responsive? Is that information available? We could not find such data in the referenced publications. This is important as the force responsiveness of TCRs is highly variable (See Liu et al., 2014, Akitsu et al, Science Advances 2024). There are many functional TCRs which operate without significant force enhancement, with suggestions that ligand density can substitute for TCR quality and these TCRs may belong to that subset. We have no expectation that such work should be required for publication, but an acknowledgment of this possibility is necessary.

Unfortunately, to the best of our knowledge, there are no investigations regarding catch- versus slipbonds with the TCR:pMHC pairs used in our study. There are only two publications which investigated single-molecule force profiles on TCR:ligand pairs in CD4+ T-cells. 1) A publication by the Garcia group investigating TCR6 and TCR11 (Sibener et al. 2018), the former one showing catch bond and the latter slip bond behavior. 2) A publication by the Zhu group (Hong et al. 2015) examining force profiles using peptide-presenting IE^k in combination with 3.L2 transgenic TCR T-cells, however they used the CD8+ variety avoiding the contribution of CD4 to the trimolecular binding complex.

Nonetheless, the 5c.c7:IE^k/MCC pair is known to be highly sensitive, with a single IE^k/MCC molecule being able to elicit a calcium response from 5c.c7 TCR-transgenic T-cells. This information has been added to the discussion (page 9). In order to test the antigen sensitivity in our experimental settings we performed titration experiments with calcium flux measurement for both T-cell sets (see in the attached figure RB1). For the AND:IE^k/MCC pair we can confirm an even higher sensitivity.

Figure RB1: Antigen sensitivity of TCR-transgenic T-cells 5c.c7 and AND. Calcium flux measurement on fluid-phase SLBs presenting fluorescently labeled IE^k/MCC in varying densities at room temperature.

2. The biophysical characterization of the tension-gauge tether system used here was thorough in the original Brenner et al., Nano Letters, 2016 publication and further elaborated and validated technically in the prior publications from this laboratory. However, the authors made statements that the length of the tether is not problematic in face of the 13 nm spacing of the immunologic synapse that are difficult to evaluate. If, in order to attain a 10 pN force, which according to Brenner, the tether elongates to 8 nm (or to 6 nm at 5 pN) according to Fig 3b therein, this does not leave much room for the pMHC or TCR proper without significant angular changes. For the pMHC with the tether inserted, we suspect that the degrees of freedom are sufficient to allow this, but given the TCR holocomplex with attendant CD3 subunits, such angular distortions may be disallowed, leading to potential torsional strain on the TCR-pMHC bond. This is, of course, speculative, but one cannot dismiss this or other unintended consequences of insertion of this rod-like, folded structure (via Brenner) out of hand. Please address this with further qualification in the discussion and if possible provide a diagrammatic illustration of such geometries either within or alongside Fig 1A-B. If the authors were able to show complete loss of the FRET fluorescence in this system, this would go a long way to convincing the reader that such geometry is attainable.

We thank the reviewer for bringing up this point. Fig. 1A has been changed to show a tilted pulling geometry as suggested. A comment to the figure legend has been added to highlight the changed geometry and the possible consequences on forces acting on the bond. The discussion of size constraints (page 8) has been updated to include this issue and to refer to the updated fig. 1A.

Unfortunately, up to now we have not found a condition where cells stretch the sensors to near zero FRET efficiency. However, the fact that forces up to 7 pN were detected with 5c.c7:H57 pairs on gel-phase, activating SLBs (fig 2A) indicates that the lower force magnitudes found with the other samples are not limited by the sensor extension.

3. The authors call out the disadvantages of the DNA-tension-gauge tether system by virtue of not just its size, but also due to its digital character. The reviewer wonders whether this digital character may be an advantage in the TCR-pMHC system. One unintended consequence of the analog system is that it will diffuse the force (by ~ 0.35 pN/nm according to Brenner et al.,) as it is stretched, altering the force application from the T cell, presumably originating with the actomyosin machinery (Ma et al 2008, Feng et al., PNAS 2017). The DNA tethers, conversely, will transmit the force until a threshold is reached, at which point the tether unfolds. This behavior is analogous to the unfolding of the TCR under force and perhaps more compatible with a system tolerized to such digital behavior. Thus, the tether adds compliance to the system beyond any compliance of the membrane itself precisely during the ramp to “optimal” forces. This may affect the generation process within the cell. Could the authors please address this point?

We agree with the reviewer that for a complete understanding of the biomechanics of this system, also the force generation by T-cells needs to be considered. However, we believe that this more complete picture i) would not change the conclusions of our manuscript, ii) would favor the spider-silk peptide over the DNA tether.

- i) *For a serial arrangement of springs, the according compliances add up linearly. In our case, this would be any potential compliance of the T-cell itself, c_{cell} , of the force sensor, c_{sensor} , and potential connections to the supported lipid bilayers, c_{SLB} , yielding $c_{\text{total}} = c_{\text{cell}} + c_{\text{sensor}} + c_{\text{SLB}}$. If the T-cell shifted the T-cell membrane via its actomyosin machinery by a distance Δl , this would amount to a force $F = \frac{\Delta l}{c_{\text{total}}}$, and, if $c_{\text{sensor}} \gg c_{\text{cell}} + c_{\text{SLB}}$, we get $F \approx \frac{\Delta l}{c_{\text{sensor}}}$. In other words, the compliance of the sensor would dominate the force magnitude. This may well be true, but does not change our conclusion that forces are largely absent in the immunological synapse. Even more so, our previous publication revealed comparably large pulling forces of ~ 7 pN in case of a single chain antibody fragment as ligand, indicating that T-cells are capable of moving the TCR by at least $\Delta l \approx F \cdot c_{\text{sensor}} \approx 2.2$ nm.*
 - ii) *Also, the DNA hairpin sensors used in (Liu et al, PNAS, 2016; Ma et al, PNAS, 2019) featured linkers, in their case of 24 nucleotides on each end of the hairpin. In addition, the contour length of each tether is ~ 8 nm, giving in total an additional element of ~ 16 nm. In comparison, in our study the FRET efficiency of ~ 0.87 indicates a dye separation of only 3.7 nm for our sensor in the collapsed state, with a peptide length of 2–3 nm which can be stretched by force application.*
4. Could the authors provide an direct statement of the error of measurement in the text for the FRET-distance values as well as the calculated force estimates? This can be from a representative experiment herein or from a prior publication. This would be helpful to the more general reader who will not want to dig through the supplementary material for such a rule of thumb when evaluating the data.

An estimation of the measurement error (less than 0.15 pN for a typical dataset of 5000 datapoints) was appended to the section “Results – Quantification of TCR-imposed molecular forces with a Molecular Force Sensor platform” (page 4). The Methods section was amended with the description of how this error was determined (page 17).

5. The single molecule FRET-based bond lifetime experiments are executed using scFV H57 bound to the TCR of interest. The lifetimes of TCR (and preTCR) bonds under force have been shown to be approximately 10-fold greater with H57 Fab addition (Das et al., 2015, Das et al., J Biol Chem 2016). This represents a potential modulation of the result as a consequence of the measurement system. This should be mentioned in the text.

Discussion on this aspect has been added in the section “Results – Global TCR:pMHC interaction lifetime is not affected by TCR-imposed mechanical forces” (pages 7–8). In essence, single-molecule tracking experiments performed in absence of H57-derived proteins yielded similar results as our single-molecule FRET experiments. While an effect of the H57-scFv cannot be ruled out, it appears far less dramatic for our TCR:pMHC pairs than for those investigated by Das et al.

6. We are unaware of documentation of direct coupling of pMHC in the membrane of APCs to cytoskeletal elements. There have been studies that have shown to motional inhibition modulated with the tail region (e.g. Edidin et al., PNAS 1994) as well as active transport of MHC proteins (e.g. Boes et al., Nature 2002). Could the authors provide relevant citations for the utilized viscosity models of the SLBs in the text? If possible, please address this directly in the introduction or early results section. Additionally, what is the potential for contribution of additional forces prior to and with MHC clustering on the APC?

The paragraph in the section “Results – Quantification of TCR-imposed molecular forces with a Molecular Force Sensor platform,” in which we discuss bilayer fluidity, has been extended (pages 3–4). In short, MFSSs on fluid-phase SLBs have a diffusion constant similar to the fast diffusing fraction of pMHCs on APCs (Platzer et al. 2023), while MFSSs on gel-phase resemble the confined fraction of pMHCs on APCs (Platzer et al. 2023).

7. In summary, the authors have presented much of their findings as a negative result. Yet, from this reviewer’s perspective, the findings on these TCRs may be confirmatory of more recent work to characterize subsets of analog or digitally sensitive TCRs with heretofore uncharacterized single TCR force profiles on T cells engaged with multiple pMHC ligands during activation. Indeed, the experiments provided evidence for differential forces in a TCR-pMHC dependent manner, though this point is not emphasized. The experimental system provides optimal measurement in a lower force regime, 0-10 pN, than is optimal for high-performing TCR (10-25 pN), and places additional geometric constraints on the binding event. It may be that the confines of the intercellular space within the immunologic synapse limits the force profiles of these bonds and this work bolsters that argument.

We thank the reviewer for sharing our excitement about our study showing intrinsic force profiles of TCR:pMHC pairs and lifetime distributions within the context of the immunological synapse. We agree with the reviewer that we did not emphasize enough the apparent differential force profiles of the investigated TCR:pMHC pairs. To rectify this issue, we amended the Discussion (page 9) comparing forces found for the different TCR:pMHC pairs. Our data do not, however, support that T-cells take advantage of dynamic catch bonds in the investigated model system as suggested (Akitsu et al. 2024). The postulated forces of 10–25 pN would have manifested in our experiments as very low FRET efficiencies, which were not observed within the limitations of our platform. Furthermore, as we investigated CD4+ T-cells, we are unable to draw conclusions about CD8+ T-cells, which were used for most catch-bond related studies. The discussion has been amended to express this more clearly (pages 9–10). Additionally, we added a discussion of findings from a DNA-based sensor (Hu et al. 2024), which in our opinion qualitatively supports our findings (while quantitatively DNA-based sensors consistently report somewhat higher forces, as we already discussed in our 2021 article (Göhring et al. 2021)).

References

- Akitsu, Aoi, Eiji Kobayashi, Yinnian Feng, Hannah M. Stephens, Kristine N. Brazin, Daniel J. Masi, Evan H. Kirkpatrick, et al. 2024. “Parsing Digital or Analog TCR Performance Through Piconewton Forces.” *Science Advances* 10 (33): eado4313. <https://doi.org/10.1126/sciadv.ado4313>.
- Göhring, Janett, Florian Kellner, Lukas Schrangl, René Platzer, Enrico Klotzsch, Hannes Stockinger, Johannes B. Huppa, and Gerhard J. Schütz. 2021. “Temporal Analysis of t-Cell Receptor-Imposed Forces via Quantitative Single Molecule FRET Measurements.” *Nature Communications* 12 (1): 2502. <https://doi.org/10.1038/s41467-021-22775-z>.
- Hong, Jinsung, Stephen P. Persaud, Stephen Horvath, Paul M. Allen, Brian D. Evavold, and Cheng Zhu. 2015. “Force-Regulated in Situ TCR–Peptide-Bound MHC Class II Kinetics Determine Functions of CD4+ t Cells.” *The Journal of Immunology* 195 (8): 3557–64. <https://doi.org/10.4049/jimmunol.1501407>.
- Hu, Yuesong, Jhordan Rogers, Yuxin Duan, Arventh Velusamy, Steven Narum, Sarah Al Abdullatif, and Khalid Salaita. 2024. “Quantifying t Cell Receptor Mechanics at Membrane Junctions Using DNA Origami Tension Sensors.” *Nature Nanotechnology* 19 (11): 1674–85. <https://doi.org/10.1038/s41565-024-01723-0>.
- Platzer, René, Joschka Hellmeier, Janett Göhring, Iago Doel Perez, Philipp Schatzlmaier, Clara Bodner, Margarete Focke-Tejkl, et al. 2023. “Monomeric Agonist Peptide/MHCII Complexes Activate t cells in an Autonomous Fashion.” *EMBO Reports* 24 (11): e57842. <https://doi.org/10.15252/embr.202357842>.
- Sibener, Leah V., Ricardo A. Fernandes, Elizabeth M. Kolawole, Catherine B. Carbone, Fan Liu, Darren McAfee, Michael E. Birnbaum, et al. 2018. “Isolation of a Structural Mechanism for Uncoupling t Cell Receptor Signaling from Peptide-MHC Binding.” *Cell* 174 (3): 672–687.e27. <https://doi.org/10.1016/j.cell.2018.06.017>.

Reviewer 2

Major points

1. the characterization of the presenting surface needs to be improved:

- the density of the ligands in the scanning vs activating condition is not measured

We apologize that this information was missing. We have added it to the section “Results – Quantification of TCR-imposed molecular forces with a Molecular Force Sensor platform” (page 4) as well as the Methods section (page 15).

- how does it rely with densities used in figure S1F?

Scanning conditions (< 0.1 molecule per μm^2) are well below the activation threshold, while activating conditions ($50\text{--}100/\mu\text{m}^2$) are profoundly above.

- is Fig S1F obtained with mobile or immobile ligands?

Fluid-phase SLBs were used. This information has been added to the legend of Fig. S1.

- following the previous questions, the terms ‘scanning’ and ‘activating’ are not properly justified and can be misleading

With the changes outlined above we hope to have clarified that we carefully checked the activation state of the T-cells experiencing the different conditions. We characterized each condition using calcium flux measurements (see fig. S1), antigen titration assays (fig. S1), as well as recruitment assays (see fig. S3d in our previous publication (Göhring et al. 2021)). We are confident that the cells encountering these surfaces are in an antigen surveilling state prior to activation on “scanning” SLBs, and respectively activate and enter immune synapse formation upon encountering “activating” SLBs.

- for example, the authors could comment on the area of the contact zone in the two cases

A comment about the contact region has been added (page 4). We do indeed see differences in footprint size between cells in activating and scanning conditions (see figure RB2).

Figure RB2: Contact area and calcium flux analysis of AND transgenic T-cells seeded on SLBs imitating activating and scanning conditions on the same experimental day. (A) Footprint size of seeded AND T-cells on gel-phase SLBs presenting activating or scanning conditions, measured by Fura-2 excitation and manual image segmentation. (B) Calcium flux analysis of AND T-cells on the respective SLBs (positive control (PC) – His-tagged IE^k/MCC , B7.1 and ICAM-1; negative control (NC) – ICAM-1 and B7-1; activating SLB – labeled and unlabeled MFS).

2. the extraction of force and their distribution is incomplete and confusing:

- what is the relation between FRET efficiency and force ?

We apologize that this information was missing. The calibration has been described in our 2021 article (Göhring et al. 2021) and is based on an article by Brenner et al. (Brenner et al. 2016). We added a short summary in the Methods (page 16) and a reference in the main text (page 3).

- how are the histograms in Fig S2 normalized ?

The histograms are normalized by the number of traces, hence the sum over all bars gives 1. This information has been added to the figure legends. We also changed the y-axis labels of the histograms to “fraction of events” hoping to improve clarity. Additionally, we now use the same y-axis limits for all corresponding histograms and probability density function estimates (e.g., all efficiency histograms have the same y-axis limits, and all force histograms as well; accordingly for figs. S3 and S5) to enable easy comparison. We express this in the respective figure legends.

- why the integral of the difference in Fig S4 is not equal to zero ?

As described in the corresponding Methods subsection (“Analysis of the high-force (=low-efficiency) fraction”), a scaled version of the “without-cells” PDF is subtracted from the “within-synapse” PDF. Since the integral of the scaled PDF is not necessarily equal to one, the integral of the difference is not necessarily zero.

- the probability densities (Fig S2) and the difference with the control of out of cells (Fig S4) are presented in two separate figures and correspond to two different sections in the results. However the same data are concerned.

Since figs. S2 and S4, in total, cover four pages and are conceptionally different, we decided to split them. Due to swapping figs. S3 and S4 (see the respective comment under Minor points below) they now appear after each other in the revised version of the manuscript. For readability, we suggest keeping them separate.

- in Fig S4 the densities are fitted before subtraction, whereas in Fig S6 they are not. It would be preferable to also avoid the fitting in S4.

In fig. S4, a fitting routine is used to determine optimal scaling factors for without-cell PDFs. This is necessary to subtract the scaled without-cell PDFs from the (unscaled) within-synapse PDFs. In no case is any kind of model fit to a PDF. To avoid confusion, we replaced the panel titles “without-cells PDF fitted” by “scaled without-cells PDF”.

- S4 and S6 also differ for the apparent choice of the y axis, which is not specified.

Y-axes are not comparable between figs. S4 and S6. Fig. S4 depicts PDF differences, fig. S6 shows CDF differences.

3. the authors claim in the discussion that one strength of their MFS is the analog force measurement. However, they introduce the “high force fraction”, from which the observations they extract seem to be qualitatively redundant with the ones obtained from the force distributions. This is especially the case since the statistical tests for the force distribution are performed with respect to zero force only and not between conditions (Fig 2A). Also, they don’t quantify what is “high-force”. The presentation would gain in being more consistent throughout the manuscript.

We thank the reviewer for their suggestions and hope we can clarify our approach: Observations from force distributions and “high-force fractions” are indeed redundant to a certain degree but still informative as the high-force fraction allows us to get insight on the frequency of force events, and the force distribution allows us to see the amplitude of these rare events.

Determining the high-force fraction is a simple mathematical procedure: For a given false-positive rate f (here: 0.05; but variation of this parameter does not strongly influence the results), a threshold was computed as the f -quantile of the FRET efficiencies recorded without cells. This procedure requires analog sensor data. Then, from experiments with cells, the fraction of FRET efficiencies below this threshold (corresponding to high forces) was computed and corrected for the expected false-positive rate f . This has been described in the “Analysis of the high-force (=low-efficiency) fraction” subsection of the Methods section, which we clarified in the revised

manuscript (page 18). We also added a reference to the Methods section from the main text (page 4). This method can be used on smaller datasets as well and is therefore suited for analysis of data split according to cell seeding time or mobility.

Analysis of the FRET efficiency distributions is more difficult. First step is unmixing the “within-synapse” distribution into stressed and unstressed (possibly unbound) sensors’ distributions. The former is unknown, the latter is expected to be similar to the “without-cells” distribution. We attempt to solve this by subtracting the scaled “without-cells” PDF from the within-synapse PDF. This works reliably only with large datasets and could therefore not be used with data split according to cell seeding time or mobility.

Testing data from cells against zero force helped us decide whether further analysis would be informative, which is why it was included in the manuscript. We did consider testing between conditions, but found little additional value for findings. However, if desired, we can add the test results.

4. Lifetime measurements:

- it would be good to superimpose the known data from the literature to the present measurements.

A reference (+ discussion prompted by reviewer #1) has been added to the section “Results – Global TCR:pMHC interaction lifetime is not affected by TCR-imposed mechanical forces” (pages 7–8).

- the temperature has an obvious effect on the lifetime. But how did the authors assess its impact on cell activation/force exertion in their experiments ?

As shown in fig. S1F, antigen discrimination by T-cells is not impaired at room temperature. We therefore expect that the recognition process is similar at room temperature and at 37°C and that our results are valid regardless of the temperature.

- the authors compare gel vs fluid phase, implicitly assuming that the vertical and lateral component of the force should affect the lifetime equivalently. They should explicit and justify better this assumption.

This has been added to the discussion (page 9). In short, since shear forces in particular have been shown to increase T-cell sensitivity, we would have expected their presence to even enhance potential force-induced effects on TCR:pMHC interaction lifetimes.

5. Discussion:

- while the title proposes an appealing hypothesis, it is not clear why the discrimination should be relevant in the so called ‘activating’ setting ?

We apologize for being unclear on this aspect. After activation, the immunological synapse can persist for hours. Continuous signaling requires ongoing ligand-dependent TCR signaling (Huppa et al. 2003). This implies that the discrimination process continues for a long time within the confines of the immunological synapse. We added this information to the section “Results – Quantification of TCR-imposed molecular forces with a Molecular Force Sensor platform” (page 4).

- differential force sharing as a function of density should be better discussed: is it not expected to have less force exerted per bond if it is shared between more bonds ? would it be also useful to consider the total force exerted by the cell for a fair comparison ?

We thank the reviewer for this suggestion. We now reflect upon load sharing for forces perpendicular as well as parallel to the T-cell membrane in the discussion (page 10), giving also an estimate of the number of affected receptor:ligand bonds for the former.

- are the cells in the same activation state in scanning and activating settings ?

In the scanning setting T-cells are predominantly not activated, while they are mostly activated in the activating setting, as verified by calcium imaging. We have described this more clearly in the revised section “Results – Quantification of TCR-imposed molecular forces with a Molecular Force Sensor platform” (see point 1 above).

Minor points

1. Fig S1: how many cells ? define the error bars

Information has been added to the figure legend.

2. Fig S3 is not called between S2 and S4.

Figs. S3 and S4 have been swapped and references adjusted.

3. Vocabulary: TCR-pMHC interaction under “strain” is used throughout while it seems to me that “stress” would be the correct word

We replaced the erroneous use of the word “strain” by “tension” as this seems to be better understood by the general readership.

4. discussion page 7: “inhibition of actin polymerization abrogates force generation”: where are these data ?

We rearranged the paragraph to clarify that this has been shown in our 2021 article (Göhring et al. 2021). We added another reference (Hu et al. 2024) which also reveals this (page 9).

5. Presentation of Fig 4C, D: i would include zero in the why axis to better emphasize the lack of difference between gel and fluid phase.

This has been changed as suggested.

References

Brenner, Michael D., Ruobo Zhou, Daniel E. Conway, Luca Lanzano, Enrico Gratton, Martin A. Schwartz, and Taekjip Ha. 2016. "Spider Silk Peptide Is a Compact, Linear Nanospring Ideal for Intracellular Tension Sensing." *Nano Lett.* 16 (3): 2096–2102. <https://doi.org/10.1021/acs.nanolett.6b00305>.

Göhring, Janett, Florian Kellner, Lukas Schrangl, René Platzer, Enrico Klotzsch, Hannes Stockinger, Johannes B. Huppa, and Gerhard J. Schütz. 2021. "Temporal Analysis of t-Cell Receptor-Imposed Forces via Quantitative Single Molecule FRET Measurements." *Nature Communications* 12 (1): 2502. <https://doi.org/10.1038/s41467-021-22775-z>.

Hu, Yuesong, Jhordan Rogers, Yuxin Duan, Arventh Velusamy, Steven Narum, Sarah Al Abdullatif, and Khalid Salaita. 2024. "Quantifying t Cell Receptor Mechanics at Membrane Junctions Using DNA Origami Tension Sensors." *Nature Nanotechnology* 19 (11): 1674–85. <https://doi.org/10.1038/s41565-024-01723-0>.

Huppa, J. B.; Gleimer, M.; Sumen, C.; Davis, M. M. Continuous T Cell Receptor Signaling Required for Synapse Maintenance and Full Effector Potential. *Nat. Immunol.* **2003**, 4 (8), 749–755. <https://doi.org/10.1038/ni951>.

Other

- *Bar plots have been replaced by box plots per request of the editor and Methods have been adapted accordingly.*
- *Data and code availability statements have been changed to reflect that our data and code to produce the figures has been deposited in an online repository.*
- *Supplementary figs. 1 and 4 were recreated using the same free and open-source tools as the other figures for better reproducibility and visual consistency*
- *Text and figures were adjusted according to the Formatting instructions and Artwork Guidelines.*

CD4+ T-cells create a stable mechanical environment for force-sensitive TCR:pMHC interactions

Manuscript Number: NCOMMS-25-05999A

1. Response to Reviewer #1

Reviewers Comment: “The authors have clarified their manuscript considerably and have provided a study that raises many questions as well as provides significant insights. My only requirement for publication is that to assist in interpreting the findings for readers, the authors should specify within the Abstract that the measurements utilized pMHC anchored to supported lipid bilayers.”

Response: We added a sentence to the abstract explaining the platform. We thank the reviewer for the constructive suggestions to improve our manuscript!

2. Response to Reviewer #2

Reviewers Comment: “The authors have satisfactorily addressed my comments.”

Response: We thank the reviewer for the constructive suggestions to improve our manuscript!